# Role of distinct fibroblast lineages and immune cells in dermal repair following UV radiation-induced tissue damage

Emanuel Rognoni[1,2]*, Georgina Goss[1], Toru Hiratsuka[1,3], Kalle H Sipilä[1], Thomas Kirk[2], Katharina I Kober[4,5], Prudence PokWai Lui[1], Victoria SK Tsang[2], Nathan J Hawkshaw[6], Suzanne M Pilkington[6], Inchul Cho[1], Niwa Ali[1,7], Lesley E Rhodes[6], Fiona M Watt[1]*

[1]Centre for Stem Cells and Regenerative Medicine, King's College London, Guy's Hospital, London, United Kingdom; [2]Centre for Endocrinology, William Harvey Research Institute, Barts and The London School of Medicine, Queen Mary University of London, London, United Kingdom; [3]Research Center for Dynamic Living Systems, Graduate School of Biostudies, Kyoto University, Kyoto, Japan; [4]Division of Signaling and Functional Genomics, German Cancer Research Center (DKFZ), Heidelberg, Germany; [5]Department of Cell and Molecular Biology, Medical Faculty Mannheim, Heidelberg University, Heidelberg, Germany; [6]Division of Musculoskeletal and Dermatological Sciences, Faculty of Biology, Medicine and Health, School of Biological Sciences, Manchester Academic Health Science Centre, The University of Manchester and Salford Royal NHS Foundation Trust, Manchester, United Kingdom; [7]The Francis Crick Institute, London, United Kingdom

*For correspondence:
e.rognoni@qmul.ac.uk (ER);
fiona.watt@kcl.ac.uk (FMW)

**Abstract** Solar ultraviolet radiation (UVR) is a major source of skin damage, resulting in inflammation, premature ageing, and cancer. While several UVR-induced changes, including extracellular matrix reorganisation and epidermal DNA damage, have been documented, the role of different fibroblast lineages and their communication with immune cells has not been explored. We show that acute and chronic UVR exposure led to selective loss of fibroblasts from the upper dermis in human and mouse skin. Lineage tracing and in vivo live imaging revealed that repair following acute UVR is predominantly mediated by papillary fibroblast proliferation and fibroblast reorganisation occurs with minimal migration. In contrast, chronic UVR exposure led to a permanent loss of papillary fibroblasts, with expansion of fibroblast membrane protrusions partially compensating for the reduction in cell number. Although UVR strongly activated Wnt signalling in skin, stimulation of fibroblast proliferation by epidermal β-catenin stabilisation did not enhance papillary dermis repair. Acute UVR triggered an infiltrate of neutrophils and T cell subpopulations and increased pro-inflammatory prostaglandin signalling in skin. Depletion of CD4- and CD8-positive cells resulted in increased papillary fibroblast depletion, which correlated with an increase in DNA damage, pro-inflammatory prostaglandins, and reduction in fibroblast proliferation. Conversely, topical COX-2 inhibition prevented fibroblast depletion and neutrophil infiltration after UVR. We conclude that loss of papillary fibroblasts is primarily induced by a deregulated inflammatory response, with infiltrating T cells supporting fibroblast survival upon UVR-induced environmental stress.

## Editor's evaluation

Your study adds important and novel information regarding how the skin responds to UV radiation and the subsequent repair and regenerative response.

## Introduction

Ultraviolet radiation (UVR) from the sun penetrates the skin and has both positive and negative impacts on human health (*Hart et al., 2019*). While UVR is essential for vitamin D synthesis, prolonged (chronic) UVR exposure contributes to the development of skin cancer (photo-carcinogenesis) and accelerates ageing (photoageing) (*Debacq-Chainiaux et al., 2012*; *Bernard et al., 2019*). UVR is a small component of solar radiation and comprises high-energy UVC (wavelength 100–280 nm), lower-energy UVB (280–315 nm), and UVA (315–400 nm) wavebands. UVC is absorbed by stratospheric ozone while UVA and UVB penetrate the skin. Chronic UVR damages the DNA, lipids, and proteins of skin cells directly (photochemical reactions) or indirectly via inflammation, reactive oxygen species (ROS) production, and matrix metalloproteinase (MMP) secretion (*Bernard et al., 2019*; *Naylor et al., 2011*; *Watson et al., 2014*; *Shih et al., 2018*).

The connective tissue of the skin, the dermis, comprises distinct layers known as the papillary, reticular, and dermal white adipose tissue (DWAT) layers (*Driskell et al., 2013*; *Rinkevich et al., 2015*). During mouse skin development, multipotent fibroblasts differentiate into distinct subpopulations (lineages) that form the different layers. These fibroblast lineages differ in location and function, and their cell identity and composition change with age (*Rognoni et al., 2016*; *Rognoni and Watt, 2018*). While papillary fibroblasts beneath the basement membrane have an active Wnt signalling signature and are required for hair follicle formation, fibroblasts in the reticular dermal layer express high levels of genes associated with extracellular matrix (ECM) and immune signalling and mediate the initial phase of wound repair (*Driskell et al., 2013*; *Rognoni et al., 2016*; *Philippeos et al., 2018*). During mouse skin development, dermal maturation is governed by a tight balance between fibroblast proliferation, quiescence, and ECM deposition. Within the first week of postnatal life, there is a coordinated switch in fibroblast behaviour from proliferative to quiescent, which is governed by ECM deposition/remodelling (*Rognoni et al., 2018*). While this quiescent state characterises postnatal skin, upon wounding different fibroblast lineages are stimulated to proliferate and migrate into the wound site (*Eming et al., 2014*). Besides depositing/remodelling ECM in the wound bed, fibroblasts are able to acquire a dermal papilla or adipocyte fate in response to distinct signals and thereby promote hair follicle and DWAT regeneration, respectively (*Gay et al., 2013*; *Lim et al., 2018*; *Plikus et al., 2017*). After tissue repair, the quiescent state of fibroblasts is restored.

UVA and UVB induce different types of photo-damage in skin. UVB penetrates the epidermis and papillary dermis, while UVA affects the full thickness of the dermis, including the subcutaneous fat (*Watson et al., 2014*; *Barnes et al., 2010*). Photoaged dermis is characterised by a loss of fibroblast density and changes in ECM organisation, including depletion of fibrillin-rich microfibers in the papillary dermis and accumulation of elastin-rich elastic fibres in the reticular dermis, which are mediated, at least in part, by MMP activity (*Naylor et al., 2011*; *Watson et al., 2014*; *Wlaschek et al., 2001*; *Scharffetter-Kochanek et al., 2000*). In addition, UVR is a potent local and systemic immune modulator, able to modify the innate and adaptive immune response (*Bernard et al., 2019*). Collectively, the activated signalling pathways and recruitment of distinct immune subsets lead to an immunosuppressive environment which supports inflammation resolution of a sunburn reaction and tissue repair but can also contribute to skin cancer.

While the consequences of UVR exposure for the epidermis, ECM, and skin-resident immune network have been widely characterised (*Debacq-Chainiaux et al., 2012*; *Bernard et al., 2019*; *Watson et al., 2014*), its short- and long-term impact on different dermal fibroblast subpopulations (lineages) is unknown. In this study, we have examined how dermal fibroblast lineages respond to acute and chronic UVB irradiation. Uncovering the UVR-induced early pathogenic processes leading to premature skin ageing and a cancer-permissive environment will pave the way for new treatment strategies that target aberrant fibroblast behaviour (*Sahai et al., 2020*). There is growing interest in the specific impact of UVB radiation on human health as ambient UVB will increase with destruction of the ozone layer (United Nations EEAP 2019 report: Environmental Effects and Interactions of Stratospheric Ozone Depletion, UV Radiation, and Climate Change, https://ozone.unep.org/science/assessment/eeap).

## Results

### Acute UVR exposure results in a transient loss of fibroblasts in the papillary dermis

We began by determining the effect of acute UVB exposure on human dermal fibroblasts (*Hawkshaw et al., 2020*). For this study, six healthy volunteers (two males, four females; mean age 44 ± 12 years) were recruited. Their buttock skin was exposed to three times their individual minimal erythema dose (MED), sufficient to induce a moderate sunburn reaction characterised by histone H2AX phosphorylation (yH2AX), a central component of the DNA damage response and repair system, and cyclobutane pyrimidine dimer (CPD) accumulation in papillary fibroblasts (*Barnes et al., 2010*). Skin biopsies were collected from irradiated skin at time points up to 14 days post-UVR and subjected to double immunofluorescence labelling for CD39 (papillary fibroblast marker) and vimentin (VM, a pan-fibroblast marker) (*Philippeos et al., 2018*; *Figure 1A*). Quantifying and plotting fibroblast density changes over time in skin sections of individual healthy volunteers revealed a loss of CD39/VM double-positive cells in the upper dermis, which was followed by a transient increase in the repair phase before returning to pre-UVR treatment level within 2 weeks.

To uncover how different dermal fibroblast subpopulations are affected by UVB irradiation, we established an acute (ac)UVR mouse model consisting of two consecutive UVB exposures (800 J/m$^2$) separated by 2 days, which induced moderate skin erythema – equivalent to a mild sunburn reaction in humans (*Figure 1B*; *Shih et al., 2018*; *Thieden et al., 2005*; *Gyöngyösi et al., 2016*). Histology revealed epidermal hyperplasia, UVB-induced angiogenesis, increased dermal immune cell infiltration, and temporary swelling of the papillary and reticular dermis at 1 day after UVB exposure (*Figure 1B*, *Figure 1—figure supplement 1A and B*). To study the effect of UVB on fibroblasts, we used the PDGFRαH2BEGFP transgenic mouse line, in which all dermal fibroblasts express nuclear GFP (*Hamilton et al., 2003*; *Collins et al., 2011*; *Figure 1C, D, E and H*). Quantification of PDGFRαH2BEGFP-positive cells showed a significant loss of upper dermal fibroblasts 1, 4, and 8 days after the second UVB exposure followed by a repair phase (*Figure 1F*), recapitulating the human in vivo findings (*Figure 1A*). The decrease in upper dermal fibroblasts correlated with an increase in DNA damage, measured by the phosphorylation of histone H2AX (yH2AX+) (*Figure 1C*), and apoptosis (cCasp3+) in dermal fibroblasts, particularly in the papillary dermis, 1 and 4 days post-UVB (*Figure 1D*). The number of immune cells (CD45+) increased markedly 1 day after irradiation and remained elevated for several days (*Figure 1E*, *Figure 1—figure supplement 1C*), accounting for the increase in total dermal cell density (DAPI+) during tissue repair (*Figure 1G*). Fibroblast proliferation increased at days 4 and 8, particularly in papillary fibroblasts, returning to normal thereafter (*Figure 1H*). No αSma+ dermal fibroblasts were present, indicating that UVB did not stimulate differentiation into myofibroblasts (*Figure 1—figure supplement 1D*). We also noted that acUVB did not induce changes in the ECM that were detectable by collagen hybridising peptide (CHP) labelling, a molecular probe that recognises the triple helix structure of immature, damaged, and remodelling interstitial collagen fibres (*Figure 1—figure supplement 1E*; *Hwang et al., 2017*).

To elucidate if distinct fibroblast subpopulations respond differently to UVR and cell stress, we reanalysed a published single-cell RNA-seq dataset of neonatal (P2) and adult (P21) mouse back skin (*Phan et al., 2020*; *Figure 1—figure supplement 2A–D*). As previously shown, neonatal fibroblasts clustered into papillary, reticular, preadipocytes, adipocytes, and other fibroblast subpopulations (*Figure 1—figure supplement 2A*), and adult fibroblasts could be divided into five major clusters. Gene Ontology (GO) analysis of differentially expressed genes between neonatal fibroblast subpopulations revealed that selected GO terms related to DNA damage, DNA repair, apoptosis, and cell stress were predominantly enriched in papillary fibroblasts, whereas expression of genes associated with GO term 'response to UV' or 'cellular response to UV' was not significantly increased in any fibroblast subpopulation at both time points (*Figure 1—figure supplement 2B and D*). Indeed, the enrichment of selected GO terms in papillary fibroblasts is due to their highly proliferative state in neonatal skin (*Rognoni et al., 2018*; *Phan et al., 2020*), rather than indicating a different susceptibility/response to UV damage. The overall enrichment of selected GO terms was much less pronounced in adult mouse skin (*Figure 1—figure supplement 2D*). Similarly, transcriptomic analysis of human sun-exposed eyelid skin (*Zou et al., 2021*) and microdissected breast skin (*Philippeos et al., 2018*) revealed that no significant GO term enrichment was associated with response to UVR or cell stress in

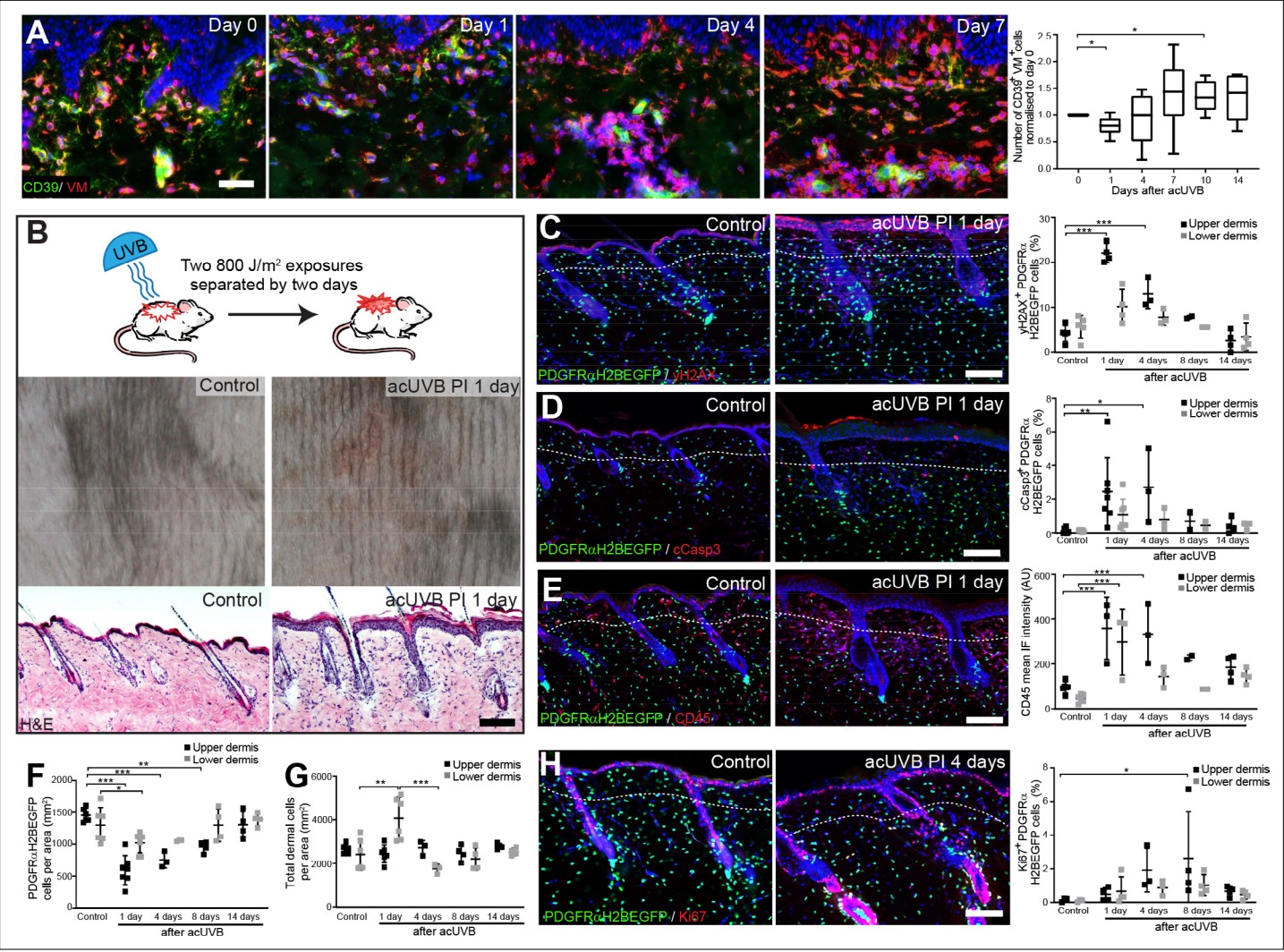

**Figure 1.** Acute UVB exposure depletes fibroblasts in the papillary dermis. (**A**) Immunostaining of human skin for CD39 (green) and vimentin (VM) (red) and quantification of double-positive cells per field of view relative to control skin at indicated time points after acute ultraviolet radiation (acUVR) exposure (n = 6 biological replicates). (**B**) Experimental design of mouse acUVR model (top panel), representative images of skin erythema (middle panel), and H&E skin section (bottom panel), showing epidermal hyperplasia and increased dermal cell density 1 day after acUVR. (**C, D**) Representative PDGFRαH2BEGFP sections (green) stained for yH2AX (**C**), and cCasp3 (**D**) (red) of control and treated skin and quantification of double-positive cells at indicated time points post-acUVR. Note that the epidermis and upper dermis show pronounced DNA damage (yH2AX+) with clusters of apoptotic cells (cCasp3+) 24 hr post-acUVB. (**E**) Immunostaining of PDGFRαH2BEGFP back skin (green) for all lymphocytes (CD45; red) and quantification of the CD45 mean fluorescence intensity at indicated time points post-UVR. (**F, G**) Quantification of dermal fibroblast density (PDGFRαH2BEGFP+) (**F**) and total dermal density (DAPI+) (**G**) 24 hr after acUVB in the upper and lower dermis. (**H**) Representative PDGFRαH2BEGFP sections (green) stained for Ki67 (red) of control and treated skin and quantification of double-positive cells at indicated time points post-acUVR. Note that the epidermis and upper dermis show increased proliferation 4 days after acUVB. Nuclei labelled with DAPI and dashed white line delineates upper and lower dermis. Scale bars, 50 μm. Data are mean ± SD. *p<0.05, **p<0.01, ***p<0.001. Source data of shown quantifications are summarised in *Figure 1—source data 1*.

The online version of this article includes the following source data and figure supplement(s) for figure 1:

**Source data 1.** Source data of quantifications represented as graphs in *Figure 1*.

**Figure supplement 1.** Dermal changes after acute ultraviolet radiation (UVR) exposure.

**Figure supplement 1—source data 1.** Source data of quantifications represented as graphs in *Figure 1—figure supplement 1*.

**Figure supplement 2.** Transcriptomic analysis of mouse and human skin fibroblasts.

**Figure supplement 2—source data 1.** Source data of Gene Ontology (GO) term analysis shown in *Figure 1—figure supplement 2*.

identified fibroblast subpopulations or in the upper and lower dermis (*Figure 1—figure supplement 2E–G*).

We conclude that in mouse and human skin acute UVB exposure results in a transient loss of papillary fibroblasts, which is associated with dermal thickening, recruitment of immune cells, and an increase in yH2AX and cCasp3-positive fibroblasts in the early acute UVB response and is followed by dermal fibroblast proliferation 4 days after UVB treatment. Our transcriptomic and GO term analysis suggests that between different fibroblast subpopulations there are only minimal intrinsic differences in UV response, DNA damage/repair, and response to cell stress.

## Chronic UVR induces long-term depletion of papillary fibroblasts and ECM changes

To test the impact of chronic UVR exposure on dermal fibroblasts, we established a chronic (ch)UVR model consisting of 800 J/m² UVB exposure twice a week for 8 weeks. This chronic UVB treatment regime in C57BL/6 mice is rather mild compared to other recent studies (*Bald et al., 2014*; *Dai et al., 2007*; *Ohkumo et al., 2006*; *Han et al., 2017*; *Meeran et al., 2009*; *Kunisada et al., 2005*) and induces a prominent tanning response (melanin deposition) in UVR-exposed back skin without disrupting the epidermal barrier (*Figure 2A*). The moderate thickening of the epidermis correlated with an increase in Ki67-positive keratinocytes (*Figure 2B*). When the skin was examined 3 days after the final UVB treatment, there was no significant difference between control and chUVB-exposed dermis in terms of proliferation (Ki67+ fibroblasts) or apoptosis (cCasp3+ fibroblasts) (*Figure 2B and C*) and no αSma-positive interfollicular fibroblasts were detected (*Figure 2—figure supplement 1A*). However, there was an increased abundance of CD45-positive cells in the upper and lower dermis and an increase in blood vessels (*Figure 2D*, *Figure 2—figure supplement 1B*). As observed in photo-aged skin (*Watson et al., 2014*), the ECM in chronically UVB-exposed skin was highly remodelled (*Figure 2E and F*). Herovici staining revealed accumulation of light blue-stained immature collagen, particularly beneath the basement membrane following UVR; in contrast, mature collagen in control skin stained pink/purple (*Figure 2E*). In addition, chUVR-treated skin showed significantly increased CHP staining, indicating that collagen fibres were damaged or actively remodelled (*Figure 2F*; *Hwang et al., 2017*). chUVR induced significant DNA damage in the epidermis and dermal fibroblasts, which was observed 1 day after the final UVB exposure. The damage was progressively repaired post-UVR exposure, as measured by staining for yH2AX+ cells (*Figure 2G*). Quantification of fibroblasts (PDGFRαH2BEGFP+) in the upper and lower dermis showed that cell density was significantly decreased in the upper dermis after the final dose of UVR and was not restored to control levels even after 30 days (*Figure 2H*). In contrast, chUVR did not affect the density of fibroblasts in the lower dermis. Total dermal cell density (DAPI+) transiently increased at 3 days post-UVR, probably due to infiltrating immune cells (*Figure 2D and I*).

We conclude that whereas acUVR leads to transient depletion of papillary fibroblasts, the effect is sustained after chronic treatment, correlating with more substantial ECM reorganisation. In contrast to acute UVR, there was minimal proliferation and apoptosis of dermal fibroblasts following chronic UVR (*Figure 2J*).

## Only fibroblasts in the papillary dermis contribute to repair of UVR damage

To understand how different fibroblast subpopulations contribute to regeneration of the papillary layer after acUVR exposure, we performed lineage tracing (*Figure 3A*). Papillary fibroblasts can be specifically labelled with Lrig1-CreER, while Dlk1-CreER marks fibroblasts in the lower dermis when Cre-mediated recombination is induced at postnatal day 0 (P0) (*Driskell et al., 2013*; *Rognoni et al., 2016*). acUVR exposure of labelled transgenics confirmed the loss of papillary fibroblasts in the upper dermis, whereas Dlk1-CreER-labelled cells in the lower dermis were not affected (*Figure 3B*). At 4 days post-UVR, papillary lineage dermal cells started to repopulate the upper dermis. However, the density of cells was significantly reduced compared to control dermis (*Figure 3B*). There were no detectable changes in the arrector pili muscle, dermal sheath, and dermal papilla fibroblasts of hair follicles upon acUVR treatment (*Figure 3B*, *Figure 1—figure supplement 1D*), suggesting that these fibroblasts did not contribute to the repair of the damaged dermis (*Rognoni et al., 2016*; *Kaushal et al., 2015*).

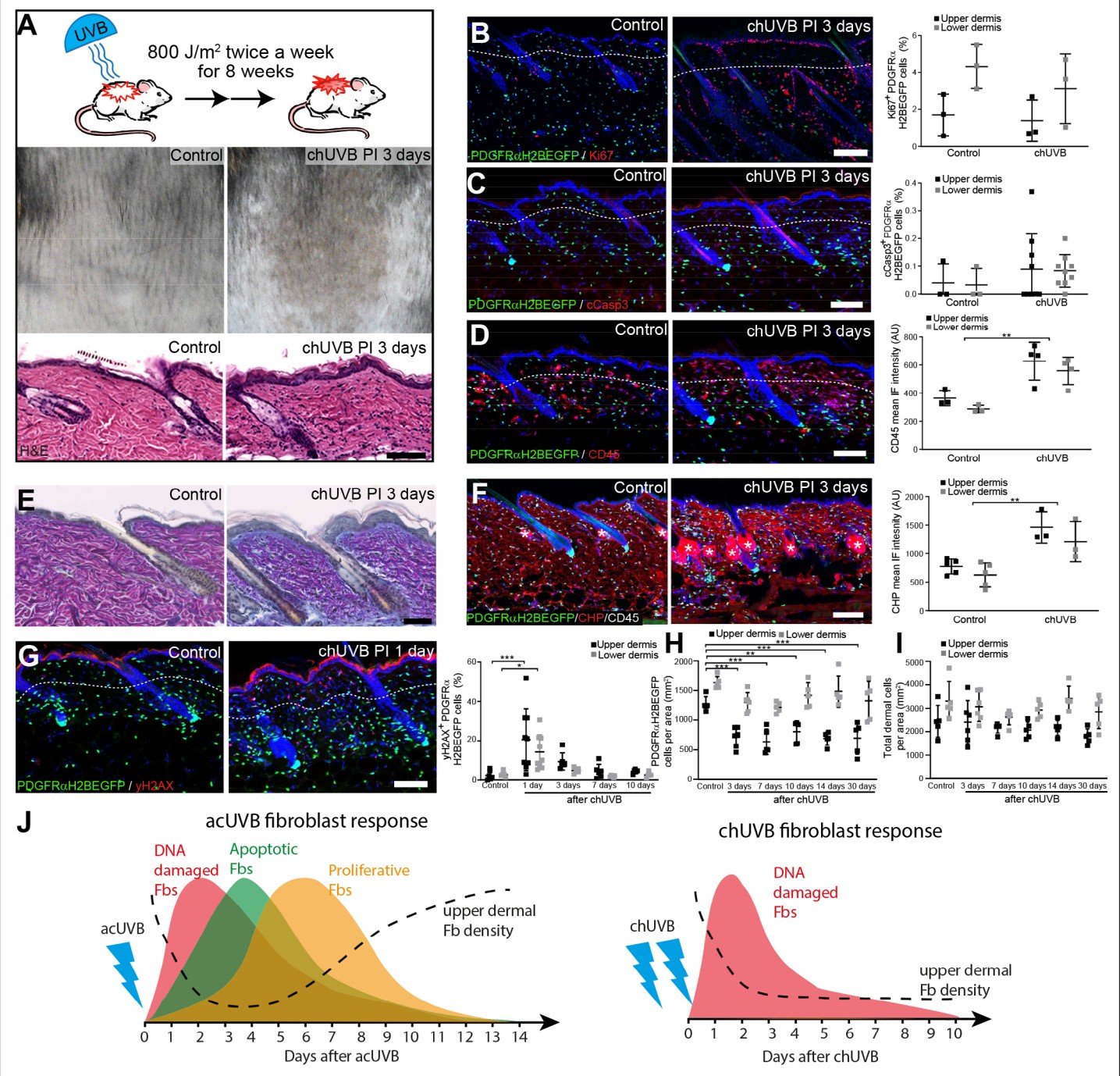

**Figure 2.** Chronic UVB irradiation leads to a permanent loss of papillary fibroblast in the upper dermis and changes in the extracellular matrix (ECM) environment. (**A**) Experimental design (top panel), representative skin tanning (middle panel), and H&E section (bottom panel), showing epidermal hyperplasia, ECM changes, and increased dermal cell density after chronic UVB (chUVB). (**B–D**) Representative PDGFRαH2BEGFP sections (green) stained for Ki67 (**B**), cCasp3 (**C**), and CD45 (**D**) (red) of control and treated skin and quantification of either double-positive cells (Ki67 and cCasp3) or mean fluorescence intensity (CD45). While lymphocytes (CD45+ cells) are increased in the dermis, pronounced proliferation (Ki67+) and apoptosis (cCasp3+) are only observed in the epidermis after chUVB. (**E**) Herovici staining of control and chUVB-exposed skin sections. Note that pink/purple staining indicates mature collagen, whereas light blue-stained collagen in chUVB skin below the basement membrane is immature and actively remodelled. (**F**) Immunofluorescence staining of control and chUVB PDGFRαH2BEGFP skin (green) for CD45 (white) and collagen (red) using the collagen hybridising peptide (CHP)-biotin probe. Mean CHP fluorescence signal was quantified, and increased CHP signal in chUVB skin indicates a more fibrillar, open, and/or damaged collagen structure. White asterisks indicate unspecific CHP staining in sebaceous glands. (**G**) Immunostaining of control and chUVR-exposed PDGFRαH2BEGFP back skin (green) for yH2AX (red) and quantification of double-positive cells at indicated time points.

*Figure 2 continued on next page*

*Figure 2 continued*

Note that the epidermis and dermis show pronounced DNA damage (yH2AX+) at 24 hr after ultraviolet radiation (UVR) which is repaired over time. (**H, I**) Quantification of dermal fibroblast (PDGFRαH2BEGFP+) (**H**) and total dermal cell density (DAPI+) (**I**) after chUVB. (**J**) Comparison of acute UVR (acUVR) and chUVR fibroblast tissue damage repair response. While acUVR induced a transient fibroblast depletion caused by DNA damage, fibroblast apoptosis, and following proliferation, chUVR led to a persistent loss of fibroblasts in the papillary dermis. Nuclei were labelled with DAPI (blue), and dashed white line delineates upper and lower dermis. Scale bars, 50 µm. Data are mean ± SD. *p<0.05, **p<0.01, ***p<0.001. Source data of shown quantifications are summarised in *Figure 2—source data 1*.

The online version of this article includes the following source data and figure supplement(s) for figure 2:

**Source data 1.** Source data of quantifications represented as graphs in *Figure 2*.

**Figure supplement 1.** Dermal changes after chronic ultraviolet radiation (UVR) exposure.

Next, we investigated how chUVR exposure impacted the papillary (Lrig1-CreER) and reticular (Dlk1-CreER) fibroblast lineages (*Figure 3C*). As in the case of acUVR exposure, Dlk1-CreER-labelled cells of the reticular dermis did not expand or contribute to tissue repair (*Figure 3D*). Lrig1-CreER-labelled cells were significantly reduced in the upper and lower dermis and showed a patchy distribution. Closer examination of papillary fibroblasts in chUVR skin revealed that, in contrast to acUVR exposure, their shapes were significantly elongated, suggesting that increased membrane protrusions may compensate for the fibroblast loss, as previously observed in aged skin (*Marsh et al., 2018*; *Figure 3E and F*, *Figure 3—figure supplement 1*).

We conclude that upon acUVR the upper dermis was replenished by papillary fibroblasts. In contrast, papillary fibroblasts in chUVR-treated skin were not replenished and instead changed shape, increasing their cell membrane protrusions (*Figure 3G*). The lower dermal lineage was unaffected by acute or chronic UVR.

## Minimal movement of fibroblasts during the UVR tissue damage response

The number of fibroblasts in the papillary dermis was significantly lower in 1, 4, and 8 days post-acUVR skin than control (non-irradiated) skin (*Figure 1F*). To explore whether the papillary dermis was depleted and repopulated via cell migration, we performed live imaging of anaesthetised PDGFRαH-2BEGFP mice 1 day and 4 days after acUVR exposure (*Figure 4A*). In each case, we recorded the movement of fibroblasts within defined fields up to 100 µm into the dermis, covering the papillary and upper reticular dermis, for 80 min. In agreement with previous measurements (*Rognoni et al., 2018*; *Marsh et al., 2018*), most fibroblasts in untreated adult skin maintained positional stability and showed minimal displacement over time (observed in three out of three imaged biological replicates) (*Figure 4—figure supplement 1A*, *Figure 4—video 1*).

Overall, we observed very little cell migration after acUVB exposure. At 1 day post-acUVB, for example, we could only find single papillary fibroblasts moving into the deeper dermis or within the horizontal plane (*Figure 4B*, top panel, *Figure 4—video 2*). Quantification of cell displacement in the horizontal and vertical directions indicated that fibroblasts were slightly more motile along the (z) axis 1 day post-irradiation compared to control skin (*Figure 4C*, *Figure 4—figure supplement 1B*). However, most fibroblasts displayed minimal displacement and the direction of the observed movement appeared heterogenous (*Figure 4C*, *Figure 4—figure supplement 1B*). At 4 days after UVR exposure, more fibroblasts showed increased random cell displacement within the horizontal and vertical dermal plane across the imaged dermis (*Figure 4B and C*, *Figure 4—figure supplement 1B*, *Figure 4—video 3*). Consistent with this, the average mean displacement speed of fibroblasts was more heterogenous at 1 day post-UVR compared to control skin and increased significantly at 4 days post-UVR (*Figure 4D*). Plotting the individual cell mean speed across the upper dermis revealed that fibroblasts with increased motility were present throughout the upper dermis at 4 days after UVR (*Figure 4E*).

We conclude that fibroblast depletion in the papillary layer in the early UVR response is not associated with cell migration, whereas at 4 days post-acUVR fibroblasts become more motile, which correlates with ECM remodelling and fibroblast redistribution (*Figure 3B and G*). The lack of directional migration indicates that fibroblast replenishment of the papillary dermis after UVR damage is

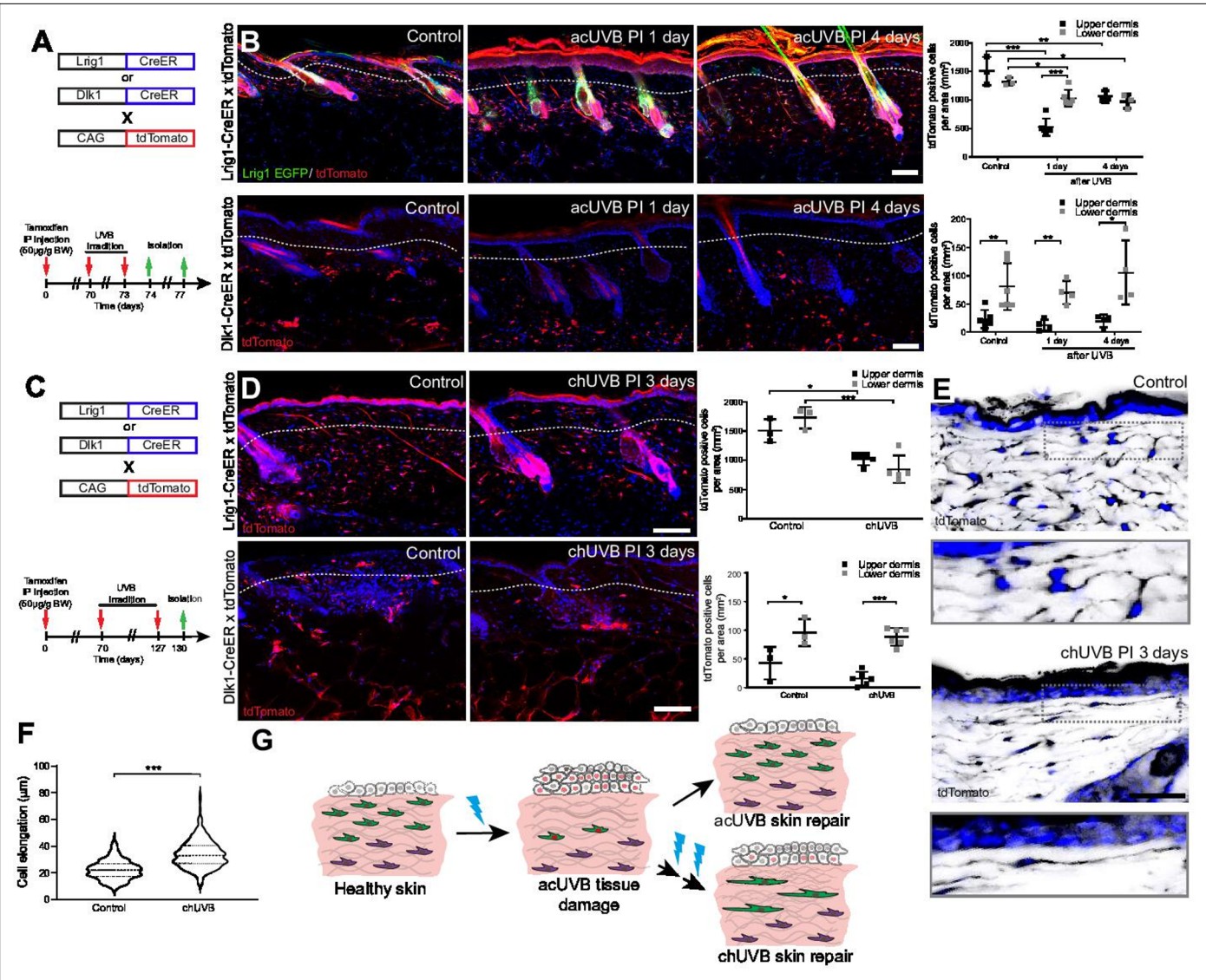

**Figure 3.** Only fibroblast lineages of the papillary dermis contribute to ultraviolet radiation (UVR)-induced tissue repair and fibroblasts in chronic UVR-exposed skin are more elongated. (**A, B**) In vivo lineage tracing of distinct dermal fibroblast populations during tissue damage repair after acute UVB (acUVB). (**A**) Experimental design shows breeding strategy and skin isolation time points to follow fibroblast lineages during tissue repair. (**B**) Representative immunofluorescence image and quantification of Lrig1-CreER × tdTomato (top panels) and Dlk1-CreER × tdTomato (lower panels) back skin of control and acUVB-exposed skin after 1 and 4 days. Quantification shows labelled cells in the upper and lower dermis at indicated time points. (**C, D**) In vivo lineage tracing of distinct dermal fibroblast populations during chUVR. (**C**) Experimental design shows breeding and lineage-tracing strategy for chronic UVB (chUVB)-exposed skin. (**D**) Immunofluorescence image and quantification of Lrig1-CreER × tdTomato (top panels) and Dlk1-CreER × tdTomato (lower panels) back skin of control and chUVB-exposed skin 3 days after last UVR exposure. Quantification shows labelled tdTomato+ cells in the upper and lower dermis. (**E, F**) Closeup of Lrig1-CreER × tdTomato lineage-traced skin section showing cytoplasmic tdTomato signal (black) (**E**) and quantification of papillary fibroblast elongation in control and chUVB-exposed skin (**F**) (n = 300 cells from four biological replicates). Boxed areas in (**E**) indicate magnified fibroblasts shown below. Note that although fibroblast density in chUVB skin is reduced (**D**), fibroblast membrane protrusions are increased. (**G**) Summary of UVR-induced tissue damage and skin regeneration after acute and prolonged (chronic) UVB exposure. In healthy skin, papillary (green) and reticular (violet) fibroblasts are quiescent. After acUVR exposure, papillary fibroblasts are depleted and epidermal and dermal cells start proliferating (red nucleus) during the tissue repair response. While fibroblast density and skin homeostasis are restored after acUVB tissue damage, repeated UVB exposure leads to a permanent loss and elongation of papillary fibroblasts and changes in the extracellular matrix (ECM) structure characteristic of aged skin. Nuclei were labelled with DAPI (blue), and dashed white line delineates upper and lower dermis. Scale bars, 50 μm. Data are mean ± SD. *p<0.05, **p<0.01, ***p<0.001. Source data of shown quantifications are summarised in *Figure 3—source data 1*.

The online version of this article includes the following source data and figure supplement(s) for figure 3:

**Source data 1.** Source data of quantifications represented as graphs in *Figure 3*.

*Figure 3 continued on next page*

*Figure 3 continued*

**Figure supplement 1.** Fibroblast shape and size at 4 days after acute UVB (acUVB) exposure in the papillary dermis.

**Figure supplement 1—source data 1.** Source data of quantifications represented as graphs in *Figure 3—figure supplement 1*.

a stochastic process similar to the fibroblast redistribution observed during dermal maturation and ageing (*Rognoni et al., 2018*).

## Activation of epidermal Wnt signalling does not enhance dermal recovery after UVR exposure

Epidermal Wnt signalling is a potent regulator of fibroblast behaviour during skin development and wound healing (*Driskell et al., 2013*; *Collins et al., 2011*; *Lichtenberger et al., 2016*). To explore how Wnt/β-catenin signalling is regulated by UVR, we subjected TOPEGFP reporter mice to acUVR. In these mice, H2BeGFP is expressed under the control of multiple Lef1/TCF binding sites, allowing nuclear GFP expression to be used as a readout of Wnt/β-catenin signalling activity (*Ferrer-Vaquer et al., 2010*). Wnt/β-catenin activity was highly induced in epidermal and dermal cells at 1 and 4 days post-acUVR (*Figure 5A*), which coincided with fibroblast DNA damage repair and proliferation (*Figure 1C and H*). During tissue repair and fibrosis, Wnt signalling has been shown to cooperate with YAP/TAZ signalling at multiple levels (*Piersma et al., 2015*; *Rognoni and Walko, 2019*). Consistent with this, dermal fibroblasts of the papillary dermis and IFE keratinocytes displayed increased nuclear YAP localisation in acute and chronic UVR-exposed skin (*Figure 5B*, *Figure 5—figure supplement 1*).

To test whether induction of fibroblast proliferation by epidermal Wnt signalling modified the response to UVR, we crossed PDGFRαH2BEGFP mice with Krt14ΔNβ-cat mice, which express stabilised β-catenin under the control of the *Krt14* promoter upon tamoxifen application (*Figure 5C*). Analysing fibroblast distribution 8 days post-UVR and epidermal β-catenin stabilisation in PDGFRαH2BEGFP × Krt14ΔNβ-cat transgenics revealed that fibroblasts predominantly proliferated and expanded around existing and ectopic hair follicles in the lower dermis (*Figure 5D–F*). However, this increased abundance of fibroblasts failed to efficiently repopulate the interfollicular dermis beneath the basement membrane and failed to restore fibroblast homeostasis and organisation in the papillary dermis.

In conclusion, although Wnt signalling is activated by UVR in the upper and lower dermis, increasing fibroblast proliferation by genetically stabilising epidermal β-catenin was not sufficient to improve fibroblast regeneration of the papillary dermis.

## Dermal fibroblast survival is supported by cutaneous T cells that become activated and expand throughout the dermis after UVR exposure

The inflammatory response to UVR is well documented; however, the impact on different fibroblast subpopulations is unclear. Neutrophils are the first immune cell type to infiltrate into the dermal region after UVB exposure (*Savage et al., 1993*; *Katiyar et al., 1999*), and this is followed by an influx of different T cell populations (*Bernard et al., 2019*). In our acUVR model, we observed an increase in CD45+ cells that persisted even after 14 days (*Figure 1E*, *Figure 1—figure supplement 1C*). The number of neutrophils increased in skin 1 day post-UVR (*Figure 6—figure supplement 1A and B*). This was followed by an increase in the abundance, proliferation, and activation of different T cell populations, specifically CD8+ cytotoxic T cells and FoxP3+ regulatory T cells (Tregs) at 5 days post-UVR. Immunofluorescence analysis and quantification of CD3, CD8, and FoxP3 labelling revealed that CD3+ T cells were depleted in the epidermis and increased in the dermis (*Figure 6A*, *Figure 6—figure supplement 1C*). Cytotoxic T cells (CD8+) that are enriched in the lower dermis in control skin were significantly increased in the upper dermis after UVR exposure (*Figure 6B*). Similarly, Tregs (FoxP3+) that are closely associated with hair follicles in homeostasis (*Ali et al., 2017*) significantly increased and expanded throughout the dermis at 3 days after acUVB exposure (*Figure 6C*). This is consistent with the observed immune cell behaviour in human skin upon UVB exposure where neutrophil infiltration is followed by an increase in accumulation, activation, and proliferation of different T cell populations (*Hawkshaw et al., 2020*; *Rijken et al., 2006*; *Rhodes et al., 2009*).

To explore the functional consequences of the increase in different T cell populations after UVR, we depleted CD4+ and CD8+ cells with specific blocking antibodies before and during acUVB exposure

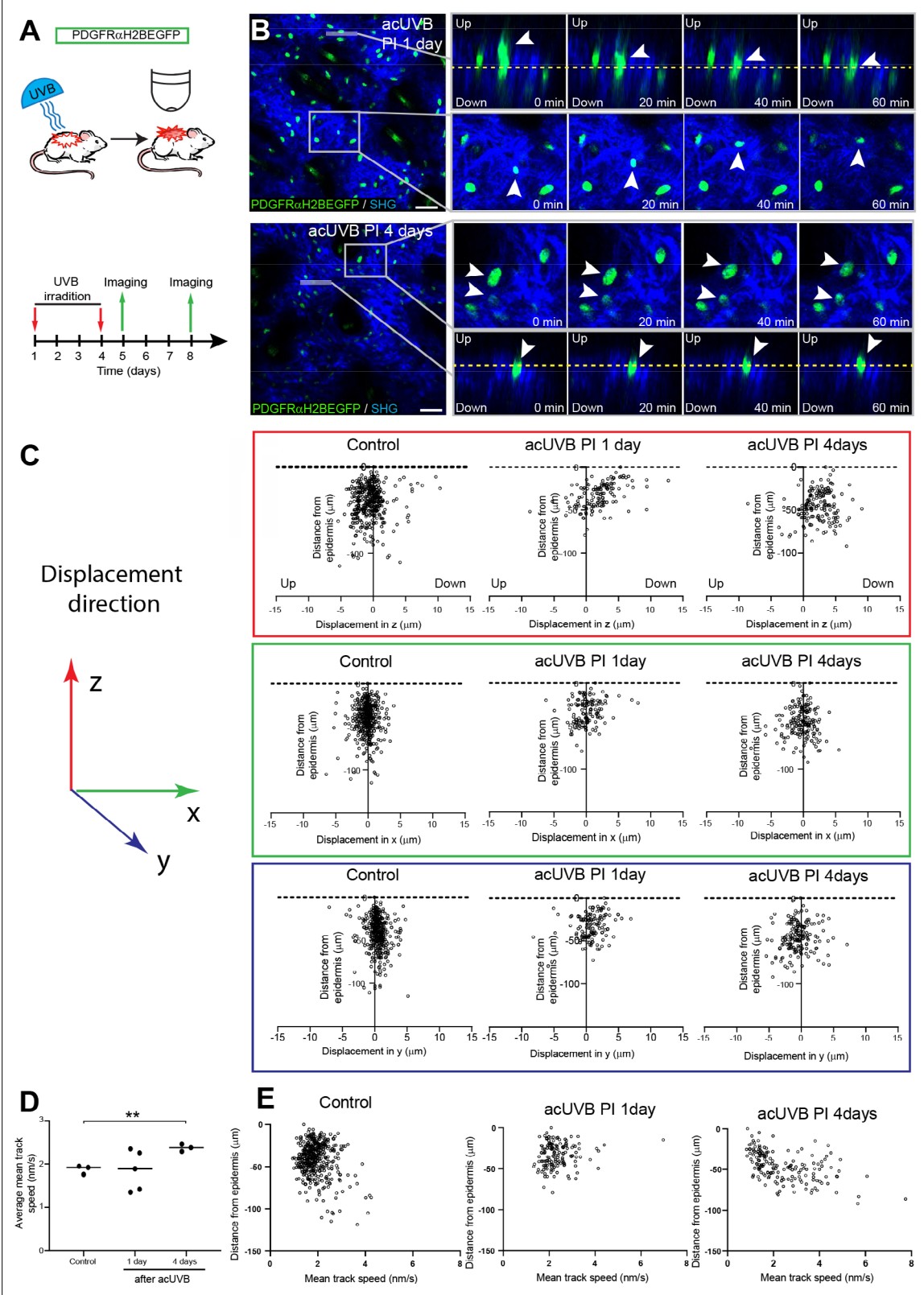

**Figure 4.** Fibroblasts in the papillary dermis become more motile during the ultraviolet radiation (UVR) tissue repair response. (**A**) Experimental design for live imaging of adult PDGFRαH2BEGFP back skin during acute UVB (acUVB)-induced tissue damage repair. (**B**) Representative time-lapse images of adult PDGFRαH2BEGFP (green) dermis 1 day (upper panel, relates to *Figure 4—video 2*) and 4 days (lower panel, relates to *Figure 4—video 3*) post-acUVB with collagen shown as second harmonic generation (SHG) in blue at indicated imaging time points. Line indicates orthogonal closeup to follow

*Figure 4 continued on next page*

*Figure 4 continued*

vertical cell displacement, and box shows fibroblast movement in the horizontal plane. Arrowheads in closeups indicate cells migrating, and dashed line is for orientation. (**C**) Scatter plots of the displacement along the indicated axis (z-, red; x-, green; y-axis, blue) of individual control and acUVB-treated cells in their relative z-location (distance from epidermis). (**D**) Average mean cell displacement speed of imaged control and acUVB-exposed back skin after 1 and 4 days. (**E**) Scatter plots of mean velocity of individual cells in their relative z-location from representative control and acUVB-treated animals after 1 and 4 days post-UVB. Scale bars, 50 μm. *p<0.05. Source data of quantifications are summarised in *Figure 4—source data 1*.

The online version of this article includes the following video, source data, and figure supplement(s) for figure 4:

**Source data 1.** Source data of quantifications represented as graphs in *Figure 4*.

**Figure supplement 1.** Live imaging of control skin and fibroblast displacement direction.

**Figure supplement 1—source data 1.** Source data of quantifications represented as graphs in *Figure 4—figure supplement 1*.

**Figure 4—video 1.** In vivo live imaging of sham-exposed back skin (control).
https://elifesciences.org/articles/71052/figures#fig4video1

**Figure 4—video 2.** In vivo live imaging of acute UVB (acUVB)-exposed back skin after 1 day (acUVB PI 1 day).
https://elifesciences.org/articles/71052/figures#fig4video2

**Figure 4—video 3.** In vivo live imaging of acute UVB (acUVB)-exposed back skin after 4 days (acUVB PI 4 days).
https://elifesciences.org/articles/71052/figures#fig4video3

(*Figure 6D*; *Ali et al., 2017*; *Grcević et al., 2000*; *Hatton et al., 2007*). Back skin from control and UVR-treated mice was isolated 24 hr after the last treatment, and flow cytometric analysis of skin and lymph nodes confirmed successful immune cell depletion (*Figure 6D*, *Figure 6—figure supplement 1D and E*). Depletion of either CD8+ or CD4+ cells significantly increased the loss of upper (papillary) fibroblasts (*Figure 6E and F*) and was associated with a significant increase in DNA damage (yH2AX + fibroblasts) and apoptosis (cCasp3+ fibroblasts) (*Figure 6G and H*) as well as a reduction in fibroblast proliferation (Ki67+) in the upper dermis but not in epidermal keratinocytes (*Figure 6I*, *Figure 6— figure supplement 1F*). Although total dermal cells (DAPI+) in the lower dermis were significantly increased, PDGFRαH2BEGFP-positive cells were unchanged in the reticular dermal layer (*Figure 6F*, *Figure 6—figure supplement 1G*). In contrast, αCD4 or αCD8 antibody treatment had no effect on fibroblasts in non-UVB-exposed skin and injections of antibodies against IgG did not change the acute UVR skin response (*Figure 6—figure supplement 1H–K*).

In summary, we observe that acUVB exposure leads to an infiltration of neutrophils that is followed by an increase and redistribution of different T cell subpopulations in the dermis. Depletion of CD4- and CD8-positive cells significantly impairs fibroblast survival and regeneration in the upper dermis after acUVB exposure by increasing DNA damage, apoptosis, and reducing fibroblast proliferation.

## COX-2 inhibition prevents dermal fibroblast loss by controlling the inflammatory response to UVR exposure

Previous reports have suggested that CD4+ T cell depletion significantly increases and prolongs the acute UVB-induced cutaneous inflammatory response (*Hatton et al., 2007*). Acute UVR exposure induced the release of pro-inflammatory prostaglandins, including prostaglandin E2 (PGE-2) in the skin, which was further increased after CD4+ and CD8+ cell depletion (*Figure 7A and B*). Cyclooxygenase-2 (COX-2) is a key enzyme for prostaglandin synthesis and can be induced in multiple cell types in response to pro-inflammatory stimuli (*Williams et al., 1999*). Immunofluorescence analysis and quantification revealed that COX-2 expression was significantly increased in the epidermis and dermis, including fibroblasts, at 1 day post-acUVB before returning to baseline 4 days after UVR exposure (*Figure 7C*). Inhibition of COX-2 following UVB irradiation has been shown to inhibit several parameters of UVR-induced acute inflammation, including vascular permeability, infiltration of neutrophils, PGE-2 production, as well as acute oxidative damage (*Wilgus et al., 2003*; *Wilgus et al., 2000*). To test whether fibroblast depletion was mainly caused by an UVR-induced inflammatory response, we treated back skin topically with the COX-2 inhibitor celecoxib immediately after UVR exposure (*Figure 7D*). Notably, COX-2 inhibition significantly inhibited fibroblast depletion in the upper dermis and neutrophil infiltration (Ly6G+) after UVR treatment (*Figure 7E–G*). Furthermore, DNA damage in dermal fibroblasts (yH2AX+) was reduced (*Figure 7H*). In contrast, celecoxib treatment did not affect fibroblast proliferation (Ki67+) and total dermal cell density (DAPI+) after acute UVR exposure (*Figure 7I and J*).

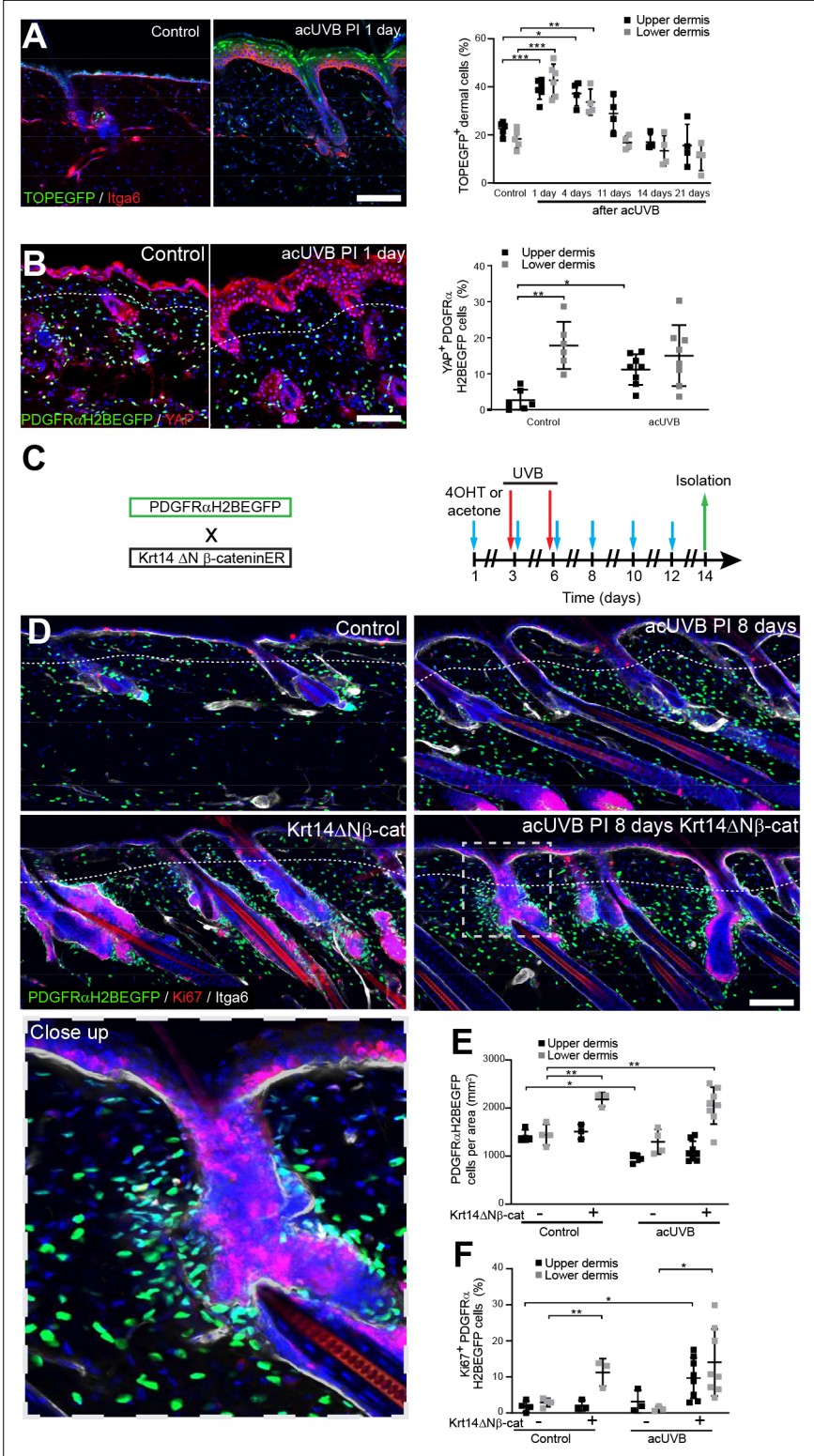

**Figure 5.** Induction of fibroblast proliferation is not sufficient to restore dermal homeostasis after ultraviolet radiation (UVR) exposure. (**A**) Representative Wnt signalling reporter (TOPEGFP) sections of control and treated skin stained for Itga6 (red). H2BEGFP (green) is expressed under the control of multiple Lef1/TCF binding sites reporting active Wnt/β-catenin signalling (***Ferrer-Vaquer et al., 2010***).Quantification of TOPEGFP-positive dermal cells in control and UVR-treated skin is shown. Note that Wnt/β-catenin signalling is increased in the epidermis as well as in the dermis. (**B**) Representative PDGFRαH2BEGFP back skin sections (green) stained for YAP (red) 1 day

*Figure 5 continued on next page*

*Figure 5 continued*

after acute UVB (acUVB) exposure. Quantification of PDGFRαH2BEGFP-positive cells with nuclear YAP in the upper and lower dermis is shown. Nuclear YAP is increased in the papillary dermis and IFE after acUVB exposure. (**C**) Experimental strategy for increasing fibroblast proliferation during acUVB damage tissue repair by stabilising epidermal β-catenin (Krt14ΔNβ-cat transgenic). (**D**) Representative PDGFRαH2BEGFP back skin sections (green) of indicated transgenics stained for Ki67 (red) 8 days post-UVR. Dashed box indicates closeup area shown in the lower panel. (**E, F**) Quantification of dermal fibroblast density (PDGFRαH2BEGFP+) (**E**) and proliferation (Ki67+ PDGFRαH2BEGFP cells) (**F**) in the indicated treatment conditions. Nuclei were labelled with DAPI (blue), and dashed white line delineates upper and lower dermis. Scale bars, 50 μm. Data are mean ± SD. *p<0.05, **p<0.01, ***p<0.001. Source data of quantifications shown are summarised in *Figure 5—source data 1*.

The online version of this article includes the following source data and figure supplement(s) for figure 5:

**Source data 1.** Source data of quantifications represented as graphs in *Figure 5*.

**Figure supplement 1.** YAP localisation in chronic UVB (chUVB)-treated skin.

**Figure supplement 1—source data 1.** Source data of quantification represented as graph in *Figure 5—figure supplement 1*.

Besides inhibiting fibroblast proliferation, collagen synthesis, migration, and differentiation into myofibroblasts, PGE-2 has been recently shown to increase fibroblast apoptosis through E prostanoid (EP)2 and EP4 receptor signalling, resulting in activation of phosphatase and tensin homologue on chromosome 10 (PTEN) and downstream inhibition of protein kinase B /AKT, an important pro-survival signal (*Huang et al., 2009*). EP4 is the major EP receptor in skin (*Regard et al., 2008*) and is expressed by dermal fibroblasts (*Joost et al., 2020*; *Sennett et al., 2015*). Immunofluorescence analysis and quantification of EP4 showed that EP4 was strongly increased in the epidermis and dermis and remained elevated in the upper dermis up to 14 days post-UVR (*Figure 7K*). In line with this, immunostaining skin sections for EP4 in the UVB-treated human skin time course revealed that EP4 expression was significantly increased in the epidermis and dermis at 1 and 4 days after acUVB exposure before returning to lower levels after 7 days in both skin compartments (*Figure 7L*). These findings suggest that the UVB-induced increase in PGE-2 level and EP4 expression influences the dermal fibroblast UVB damage response and survival (*Figure 7M*).

In summary, we have shown that the loss of fibroblasts in the upper dermis is primarily induced by an UVR-induced inflammatory response involving PGE-2 and EP4 signalling that can be supressed by COX-2 inhibition. Our data suggest that infiltrating/activated T cells support fibroblast survival and regeneration following UVR-induced environmental stress by controlling the inflammatory response to UVR exposure (*Figure 7M*).

## Discussion

In this study, we have elucidated the short- and long-term impacts of UVR exposure on different fibroblast lineages in the skin. We reveal that physiological doses of UVR are sufficient to severely deplete papillary fibroblasts in human and mouse skin, and that fibroblast survival is influenced by cutaneous T cells and PGE-2/EP4 receptor signalling. Our immunofluorescence, lineage tracing, and in vivo live imaging results showed that the loss of papillary fibroblasts is primarily due to apoptosis rather than movement of papillary fibroblasts into the deeper dermis. After acute UVR fibroblasts start proliferating, increase motility and restore tissue density. In contrast, prolonged exposure to UVR prevented repopulation of fibroblasts in the upper dermis even after 30 days post-UVR (*Figure 2H*). These observations are in line with previous studies of chronic UVB irradiation that reported a reduced cell density in the papillary dermis even 200 days after final UVB exposure (*Dai et al., 2007*). Loss of the papillary lineage is associated with premature skin ageing, reduced regeneration, and a profibrotic environment (*Driskell et al., 2013*; *Rognoni et al., 2016*; *Phan et al., 2020*; *Lichtenberger et al., 2016*).

We and others have recently shown how different fibroblast lineages contribute to dermal architecture and have explored their tissue-scale behaviour in development and skin regeneration (*Rognoni et al., 2018*; *Marsh et al., 2018*; *Jiang et al., 2018*). Deregulation of these complex processes is associated with several skin pathologies, including fibrosis, chronic wounds, and cancer. Comparison of the fibroblast lineage response during repair of full thickness wounds and UVR-induced tissue damage reveals several differences. While both forms of tissue damage induce a pronounced inflammatory

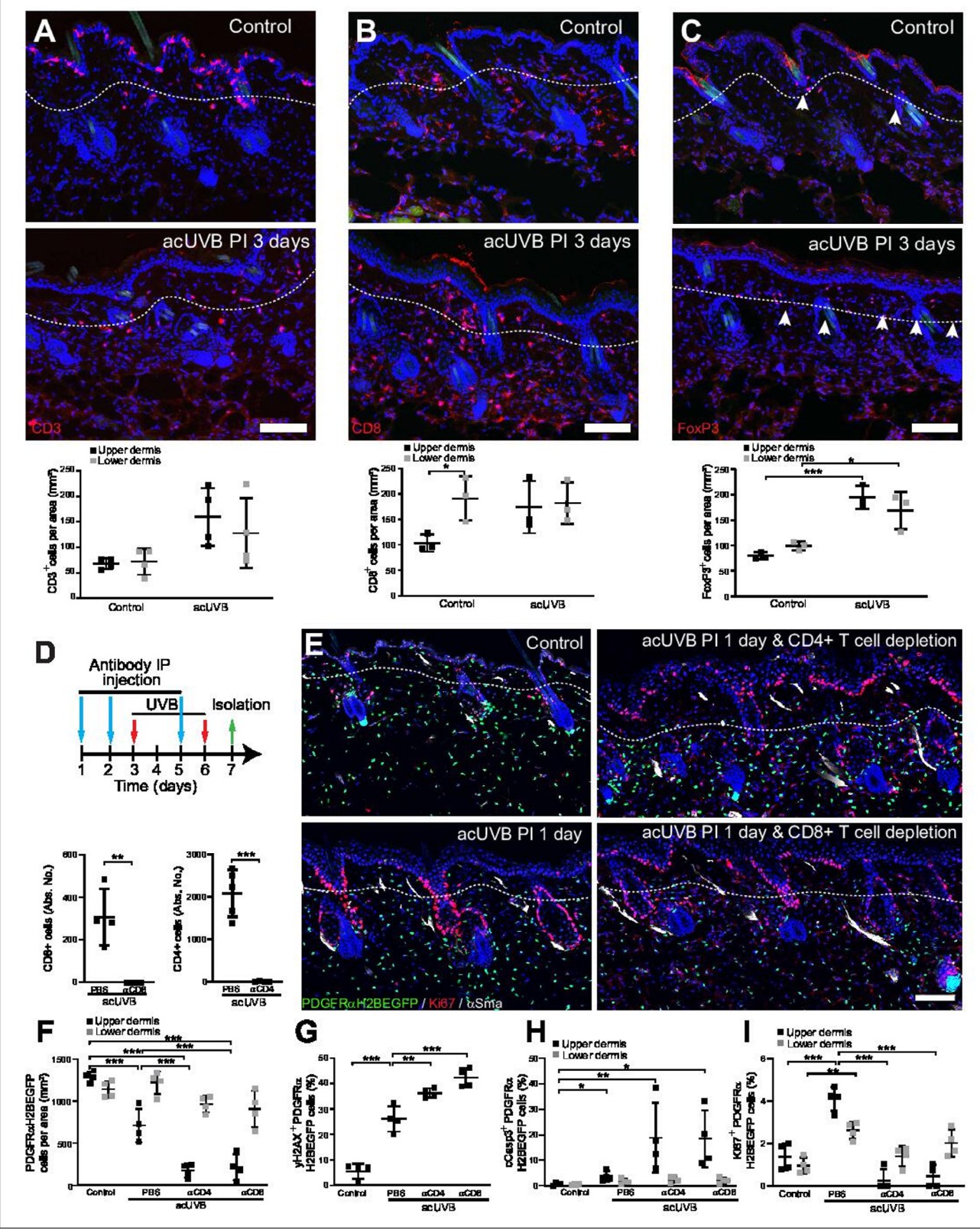

**Figure 6.** Cutaneous T cells redistribute in response to acute UVB (acUVB) exposure and influence dermal fibroblast survival. (**A–C**) Immunostaining of CD3 (CD3+ T cells) (**A**), CD8 (cytotoxic T cells) (**B**), and FoxP3+ (Tregs) (**C**), in red. Note the pronounced depletion of CD3+ T cells in the epidermis and redistribution of activated Tregs (white arrow heads) and cytotoxic T cells in the interfollicular dermis 3 days after acute ultraviolet radiation (acUVR). (**D–I**) CD4- and CD8-positive cell depletion increased fibroblast loss in the upper dermis after acUVB. (**D**) Experimental strategy for antibody-

*Figure 6 continued on next page*

*Figure 6 continued*

based immune cell depletion during acUVB (blue arrow, antibody injection; red arrow, UVB; green arrow, skin isolation) (top panel). Antibody depletion was assessed by FACS analysis of cutaneous CD4- and CD8-positive cells. Absolute number quantifications are for 6 cm² (bottom panels). (**E**) Representative immunostaining of PDGFRαH2BEGFP back skin (green) for Ki67 (red) and αSma (white). (**F–I**) Quantification of dermal fibroblast density (PDGFRαH2BEGFP+) (**F**), DNA damage (yH2AX + PDGFRαH2BEGFP cells) (**G**), apoptosis (cCasp3 + PDGFRαH2BEGFP cells) (**H**), and proliferation (Ki67+ PDGFRαH2BEGFP cells) (**I**) after acUVB and indicated treatment conditions. Nuclei were labelled with DAPI (blue), and dashed white line delineates upper and lower dermis. Scale bar, 50 μm. IP, intraperitoneal injection. Data are mean ± SD. *p<0.05, **p<0.01, ***p<0.001. Source data of quantifications are summarised in *Figure 6—source data 1*.

The online version of this article includes the following source data and figure supplement(s) for figure 6:

**Source data 1.** Source data of quantifications represented as graphs in *Figure 6*.

**Figure supplement 1.** Immune cell infiltration after acute UVB (acUVB) exposure and during tissue repair.

**Figure supplement 1—source data 1.** Source data of quantifications represented as graphs in *Figure 6—figure supplement 1*.

response and activation of Wnt/β-catenin and YAP/TAZ signalling in the epidermis and dermis, UVR-induced tissue damage is repaired with minimal fibroblast proliferation, migration, and myofibroblast differentiation. We recently showed that ECM is a potent regulator of fibroblast behaviour and inhibits proliferation during skin development and regeneration (*Rognoni et al., 2018*). The inhibitory signal of the ECM can be partly overcome by overexpression of epidermal β-catenin, which induces the expression of several fibroblast growth factors (*Lichtenberger et al., 2016*). However, the induction of proliferation especially in the lower dermis was not sufficient to restore fibroblast organisation in UVR-damaged skin, which could be due to a lack of directional migration. Consistent with this, our lineage-tracing and in vivo live imaging experiments revealed that only fibroblasts in the upper dermis contribute to tissue repair (*Figure 3G*). While in the early UVR response (1 day post-UVR) fibroblast migration is limited to single cells, papillary fibroblasts become more motile at 4 days post-UVR, which could be due to ECM remodelling. In support of this concept, during skin homeostasis most dermal fibroblasts are stationary, yet active random fibroblast migration has been observed close to growing hair follicles where the surrounding ECM is extensively remodelled (*Marsh et al., 2018*). In contrast, upon full-thickness wounding, we and others have shown that fibroblasts start migrating towards the wound where they randomly distribute and expand during the early wound repair phase (*Rognoni et al., 2018*; *Jiang et al., 2020*). During wound healing, chemoattractants such as platelet-derived growth factor (PDGF) are key regulators of fibroblast chemotaxis (*Melvin et al., 2011*); however, the intrinsic and extrinsic signals controlling fibroblast migration after UVR exposure remain unclear.

Recent laser or genetic ablation experiments have revealed that loss of dermal fibroblasts is repaired through a mixture of proliferation/migration and reorganisation of the plasma membrane network (*Marsh et al., 2018*). Our data indicate that a similar mechanism may apply during repair of UVR-induced tissue damage. In contrast to acUVR tissue damage, in chUVR skin the decreased fibroblast density persisted in the papillary dermis and surviving fibroblasts were significantly elongated; in addition, fibroblast loss was compensated by an increased membrane network of surrounding fibroblasts. This is in line with the observation of Marsh et al. that the progressive loss of fibroblasts during skin ageing is balanced by increasing membrane protrusions rather than fibroblast proliferation or migration (*Marsh et al., 2018*). Our data indicate that repeated UVB tissue damage accelerates this process (photoageing).

Resident immune cells are not only essential for skin barrier function, pathogen defence, and wound healing but also provide essential signals for hair follicle growth and skin regeneration (*Gay et al., 2013*; *Ali et al., 2017*; *Nosbaum et al., 2016*). Here we identify an additional function, that of promoting the survival of dermal fibroblasts during environmental stress. In line with previously published reports, we found that upon UVR exposure neutrophils were first recruited to the UV-exposed site (*Hawkshaw et al., 2020*; *Bald et al., 2014*; *Wilgus et al., 2000*). Neutrophil-derived reactive oxidants are potent mediators of UVB-induced tissue damage and tumorigenesis because of their cytotoxicity and immunosuppression (*Savage et al., 1993*; *Katiyar et al., 1999*). This was followed by infiltration of different types of T cells; in particular, Tregs became highly activated and proliferative (*Figure 6*, *Figure 6—figure supplement 1*), which is suggested to inhibit the UVR-induced inflammatory response (*Bernard et al., 2019*). While in homeostatic conditions Tregs are predominantly located around hair follicles (*Ali et al., 2017*), their expansion throughout the interfollicular dermis was evident upon UVR exposure, and this could potentially promote an immunosuppressive

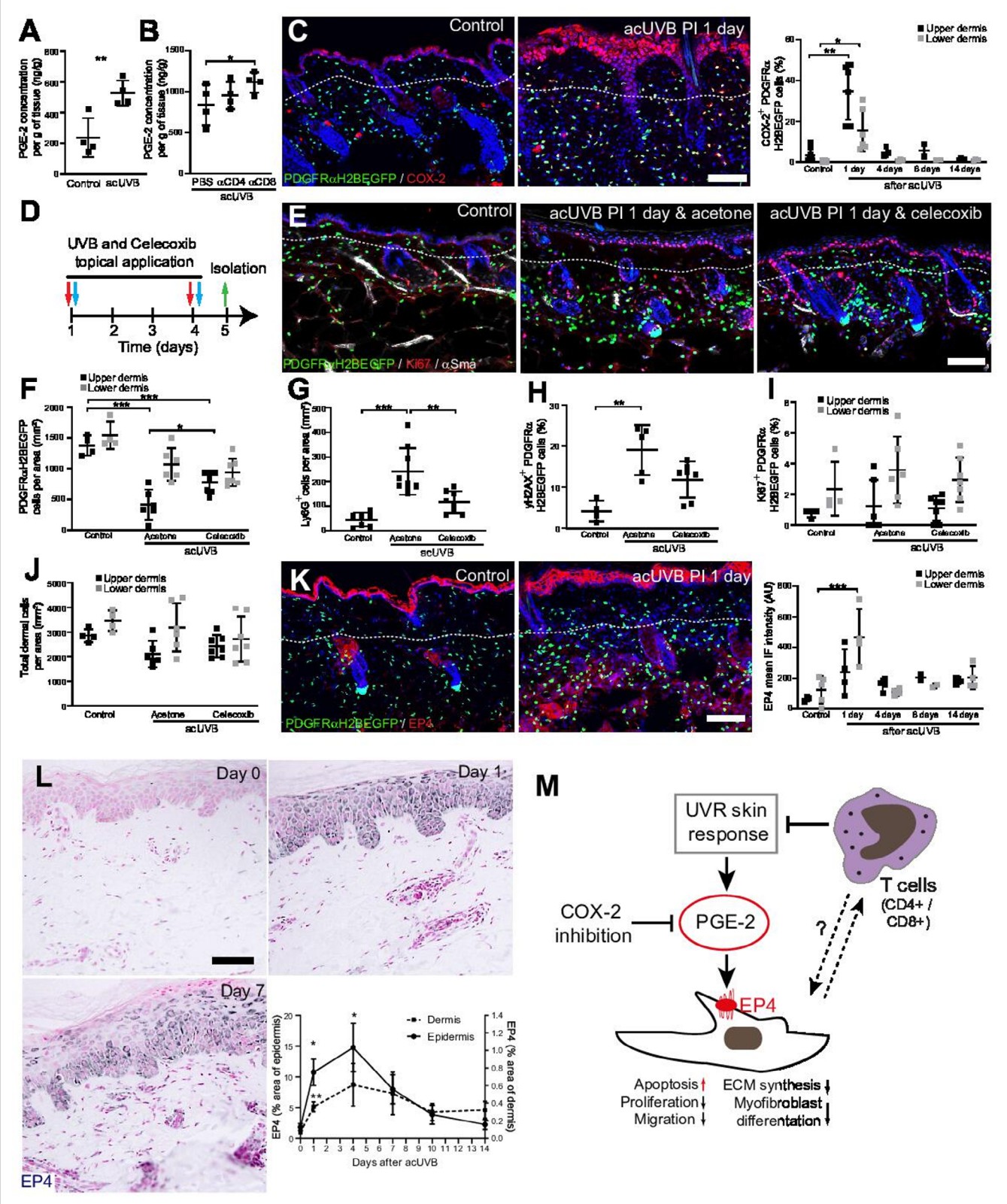

**Figure 7.** Inhibition of ultraviolet radiation (UVR)-induced inflammation increases fibroblast survival in the skin. (**A, B**) Prostaglandin E2 (PGE-2) skin concentration 24 hr after acute UVB (acUVB) exposure (**A**) and in combination with CD4+ and CD8+ cell depletion (**B**). Note that antibody depletion of CD4+ and CD8+ cells further increases the PGE-2 concentration in acUVB-treated skin. (**C**) Representative PDGFRαH2BEGFP sections (green) stained for COX-2 (red) of control and treated skin and quantification of double-positive cells at the indicated time points post-acUVR. Note that the

*Figure 7 continued on next page*

Figure 7 continued

epidermis and dermis show pronounced increase in COX-2 expression 24 hr after acUVB exposure. (D–J) COX-2 inhibition decreased fibroblast loss. (D) Experimental design for topical treatment with celecoxib (COX-2 inhibition) immediately after acUVB exposure. (E) Representative immunostaining of PDGFRαH2BEGFP back skin (green) for Ki67 (red) and αSma (white) under the indicated treatment conditions. (F–J) Quantification of dermal fibroblast density (PDGFRαH2BEGFP+) (F), neutrophil infiltration (Ly6G+) (G), DNA damage (yH2AX+ PDGFRαH2BEGFP cells) (H), fibroblast proliferation (Ki67+ PDGFRαH2BEGFP) (I), and total dermal cells (DAPI+) (J) under the indicated experimental conditions. (K) Representative PDGFRαH2BEGFP sections (green) stained for EP4 (red) and quantification of the EP4 mean fluorescence intensity at the indicated time points post-UVR. (L) Immunostaining of human skin for EP4 receptor and quantification of EP4 in the epidermal and dermal areas per field of view at the indicated time points after acUVB exposure (n = 13 biological replicates). (M) Model of PGE-2-EP4 signalling in dermal fibroblasts after UVR exposure showing the influence on tissue damage response and survival in concert with T cells. Data are mean ± SD except (L), is ± SEM. *p<0.05, **p<0.01, ***p<0.001. Nuclei were labelled with DAPI (blue), and dashed white line delineates upper and lower dermis. Scale bars, 50 µm. Source data of quantifications are summarised in *Figure 7—source data 1*.

The online version of this article includes the following source data for figure 7:

**Source data 1.** Source data of quantifications represented as graphs in *Figure 7*.

environment. Whether there is also direct cross-talk between T cells and dermal fibroblasts during the UVR tissue damage response is currently unclear. A recent study in human skin has identified CD4+ GATA3+ and CD8+ GATA3+ T cells as the predominant T cell populations in UVR-induced inflammation, and these are therefore likely to contribute to tissue resolution via dermal communication (*Hawkshaw et al., 2020*).

Production of multiple pro-inflammatory prostaglandins, including PGE-2, is promoted following UVR as a result of arachidonic acid release by phospholipases and by induction of COX-2 expression in various skin cells (*Rhodes et al., 2009*; *Fuller, 2019*; *Athar et al., 2001*). Here we show that PGE-2 and COX-2 are both significantly elevated immediately after UVR exposure and PGE-2 levels are further increased after T cell depletion. In line with our mouse data, it has been reported that following acUVB exposure of human skin pro-inflammatory prostaglandins including PGE-2 peak at 24 hr before normalising after 4 days (*Hawkshaw et al., 2020*; *Rhodes et al., 2009*). PGE-2 governs diverse biological functions that are mediated by signalling through four distinct E-type prostanoid (EP) receptors, EP1–4 (*Konya et al., 2013*). The major EP receptor in skin is EP4, which is a Gs-coupled receptor regulating cAMP/PKA, MEK/ERK1/2, NF-ßB, and PI3K/ERK/Akt signalling; these pathways are important for cell survival, proliferation, migration, differentiation, angiogenesis, and inflammation. In fibroblasts, PGE-2 binding to EP4 has been shown to increase PTEN activity and Fas expression and decrease survivin expression, thereby promoting apoptosis (*Huang et al., 2009*). In line with these observations, the increased PGE-2 levels and EP4 expression in the dermis in the early UVR response coincide with the observed fibroblast depletion at 1 and 4 days after UVB exposure (*Figure 7M*).

In support of the concept that prostaglandin signalling influences fibroblast survival, specific inhibition of COX-2 immediately after UVB exposure significantly reduced fibroblast depletion and neutrophil recruitment. This is consistent with the observation that the COX enzyme inhibitor aspirin (acetylsalicylic acid) efficiently protects keratinocytes and melanocytes from acute UVB-induced DNA damage by decreasing cutaneous inflammation and PGE-2 levels in skin (*Rahman et al., 2021*). Based on our findings, aspirin may also enhance dermal fibroblast survival and regeneration upon UVR-induced environmental stress. Whether the anti-inflammatory activities of aspirin could help prevent fibroblast-associated changes in photoaged skin warrants investigation in the future.

In summary, we have shown that papillary fibroblasts repair the dermis following UVR with minimal migration and that their survival is influenced by tissue resident T cells controlling UVR-induced inflammation. Specific inhibition of COX-2 reduces dermal fibroblast loss which may have therapeutic applications in the treatment of UVR-induced skin damages and photoageing. Our findings could also be relevant to stromal changes in skin cancer, in which fibroblast senescence is linked to age-associated cancer risk (*Lewis et al., 2010*; *Procopio et al., 2015*).

## Materials and methods
### Human volunteer and UVR time-course analysis
Ethical approval was granted by the Greater Manchester North NHS research ethics committee (ref: 11/NW/0567) for the studies presented in *Figures 1 and 7*. Details of the time-course analysis of

UVR-challenged human skin have been reported previously (*Hawkshaw et al., 2020*). Briefly, this study was conducted at the Photobiology Unit, Salford Royal NHS Foundation Trust, Greater Manchester, UK, and involved healthy volunteers aged between 18 and 60 years. All were white Caucasian, of skin phototypes I–III according to the Fitzpatrick skin phototyping scale. In each case, photo-protected upper buttock skin received a single dose of three times the MED using a UVB lamp (Waldmann 236B, peak 313 nm, 280–400 nm) to separate sites on five different days. This allowed collection of skin samples at 1, 4, 7, 10, and 14 days post-UVR, in addition to unirradiated skin. Erythema measurements and 5 mm skin punch biopsies were taken at the end of the time-course experiment, as described (*Hawkshaw et al., 2020*). Skin biopsies were bisected, with half snap frozen in optimal cutting medium and half formalin fixed and paraffin embedded. All volunteers provided written informed consent in accordance with the Declaration of Helsinki principles.

## Transgenic mice

All animal experiments were subject to local ethical approval and performed under the terms of a UK government Home Office licence (PPL 70/8474 or PP0313918). All mice were outbred on a C57BL/6 background, and male and female mice were used in experiments that included PDGFRαH2BEGFP (*Hamilton et al., 2003*), Lrig1-CreERt2-IRES-GFP (Lrig1-CreER) (*Page et al., 2013*), Dlk1-CreERt2 (Dlk1-CreER) (*Driskell et al., 2013*), Krt14ΔNβ-cateninER (Krt14ΔNβ-cat) (*Lo Celso et al., 2004*), ROSAfl-stopfl-tdTomato (Jackson Laboratories, 007905), and TCF/Lef:H2B-GFP (TOPEGFP) (*Ferrer-Vaquer et al., 2010*) mice. Animals were sacrificed by $CO_2$ asphyxiation or cervical dislocation. All efforts were made to minimise suffering for mice. For lineage tracing, transgenic reporter mice were crossed with the indicated CreER line and ER was induced by injection with 10 µl tamoxifen (50 µg/g body weight; Sigma-Aldrich) intraperitoneally in newborn mice (P0), when Dlk1 and Lrig1 are highly expressed in dermal fibroblasts (*Driskell et al., 2013*; *Rognoni et al., 2016*). Tamoxifen for injection was dissolved in corn oil (5 mg/ml) by intermittent sonication at 37°C for 30 min. For epidermal β-catenin stabilisation acUVR experiments, central back skin of Krt14ΔNβ-cat × PDGFRαH2BEGFP transgenics was clipped and treated topically with 100 µl 4-hydroxytamoxifen (4OHT) (2 mg/ml dissolved in acetone; Sigma-Aldrich) every second or third day for a total of six applications before and after sham or acUVB exposure (see experimental design in *Figure 5C*). Tissue was collected at the indicated time points, briefly fixed with 4% paraformaldehyde/PBS (10 min at room temperature), and embedded into optimal cutting temperature (OCT) compound or fixed in 4% paraformaldehyde/PBS overnight at 4°C for paraffin embedding as previously described (*Kober et al., 2018*).

## UVR acute and chronic mouse models including COX-2 inhibition and immune cell depletion

For the in vivo UVR treatments, a UVR system (Tyler Research UV-2) was used which has a cascade-phosphor UV generator lamp (TL 20 W/12 RS SLV, Philips) with a sharp 310 nm peak output (65% of UVR falls within 20 nm half bandwidth). Thus the generated UVR is highly enriched for UVB which penetrates the epidermis and upper dermis (*Watson et al., 2014*). Comparison of different mouse strains has revealed that the C57BL/6 background most closely mimics human skin UVB response (*Gyöngyösi et al., 2016*; *Sharma et al., 2011*). For UVR exposure, mice were restrained in a custom-made mouse restrainers which only exposed a defined (2 cm × 3 cm) central back skin area to UVR. The UVB dose used for the acute (two consecutive exposures separated by 2 days and isolation at the indicated time points after second treatment) and chronic (twice a week for 8 weeks and isolation at the indicated time points after last exposure) models (800 J/m²) has been shown to correspond closely to the clinically relevant UVB dose in C57BL/6 mice that induces a detectable skin reaction (erythema/oedema) (*Gyöngyösi et al., 2016*).

For immune cell depletion, in vivo anti-CD4 (clone GK1.5, 400 µg per injection in 100 µl PBS), anti-CD8 (clone 2.43, 400 µg per injection in 100 µl PBS) and anti-IgG control (clone LTF-2, 400 µg per injection in 100 µl PBS), all purchased from BioXCell (West Lebanon, NH), were administered intraperitoneally three times before and during acUVR exposure (see experimental design in *Figure 6D*). Back skin and lymph nodes were collected 24 hr after the last treatment and analysed by flow cytometry and immunofluorescence.

For COX-2 inhibition during acUVR, mice were divided into control and UVR-treated groups which were either treated topically with vehicle (200 µl acetone) or 500 µg of celecoxib (Selleckchem, S1261)

dissolved in acetone (200 µl) immediately after sham or UVR exposure (800 J/m$^2$) (see experimental design in *Figure 7D*). Mice were killed 24 hr after the second UVB treatment, and back skin was analysed as described above.

In all UVR experiments, 10- to 20-week-old male and female mice were randomised in the different experimental groups and the hair of the central back skin was clipped 24 hr prior to sham or UVR exposure. Back skin with hair follicles not in telogen (hair growth resting phase) at the beginning of the experiment was excluded. During the UVR procedures, mice were housed in small groups (≤3) to minimise the risk of fighting, and skin with signs of scratching was not included in the analysis.

## Tissue digestion and flow cytometry analysis

Preparation of single-cell suspensions for flow cytometry was performed as previously described (*Ali et al., 2017*). Briefly, isolation of cells from skin draining lymph nodes (axillary, brachial, and inguinal lymph nodes) for flow cytometry was performed by mashing tissue over 70 µm sterile filters. For isolation of skin cells, mouse dorsal skin was minced finely, resuspended in 3 ml of digestion mix (composed of 2 mg/ml collagenase XI [Sigma-Aldrich, C7657], 0.5 mg/ml hyaluronidase [Sigma-Aldrich, H4272], and 0.1 mg/ml DNase [Sigma-Aldrich, DN-25] in 10% foetal bovine serum, 1% Pen/Strep, 1 mM Na-pyruvate, 1% HEPES, 1% non-essential amino acid, 0.5% 2-mercaptoethanol in RPMI-1640 [+L glut] medium), and digested for 45 min at 37°C 255 rpm. The digestion mix was then resuspended in 20 ml of RPMI/HEPES/P-S/FCS media and passed through a 100 µm and 40 µm cell strainer before centrifugation at 1800 rpm for 4 min at 4°C. Cell pellets were resuspended in 1 ml of FACS buffer (2% foetal calf serum, 1 mM EDTA in PBS) for cell counting with an automated cell counter (NucleoCounter NC-200, Chemometec) to calculate absolute cell numbers. Following isolation from the tissue, cells were stained with surface antibodies and a live dead marker (Ghost Dye Violet 510) (see *Supplementary file 1*) for 20 min on ice. All samples were run on Fortessa 2 (BD Biosciences) at the KCL BRC Flow Cytometry Core, which was standardised using SPHERO Rainbow calibration particle, 8 peaks (BD Biosciences, 559123). For compensation, UltraComp eBeads (Thermo Fisher, 01-2222-42) were stained for each surface and intracellular antibody following the same procedure as cell staining. ArC Amine Reactive Compensation Bead Kit (Thermo Fisher, A10346) were used for Ghost Dye Live/Dead stain. All gating and data analysis were performed using FlowJo v10, while statistics were calculated using GraphPad Prism 9.

## Histochemical and immunostaining

H&E and Herovici staining of 8-µm-thick paraffin mouse skin sections was processed as previously described (*Kober et al., 2018*), and sections were mounted in DPX mounting medium (Sigma-Aldrich).

For immunostaining, mouse tissue samples were embedded in OCT compound (Life Technologies) prior to sectioning. For thin section stains, cryosections of 14 µm thickness were fixed with 4% paraformaldehyde/PBS (10 min at room temperature), permeabilised with 0.1% Triton X-100/PBS (10 min at room temperature), blocked with 5% BSA/PBS (1 hr at room temperature), and stained with the following primary antibodies to VM (1:500; Cell Signaling, #5741), Ki67 (1:500; Abcam, ab16667, and Invitrogen, clone SolA15), αSma (1:500; Abcam, ab5694), γH2AX (1:500; Abcam, ab81299), COX-2 (1:500; Abcam, ab15191), EP4 (1:200 Bioss, BS-8538R), YAP (1:500; Cell Signaling, #14074), cCasp3 (1:500; Cell Signaling, #9661), CD45 (1:200; eBioscience, clone 30-F11), CD31 (1:200; eBioscience, clone 390), CD49f (1:500; BioLegend, clone GoH3), CD3 (1:200; BioLegend, clone 17A2), CD8 (1:200; BioLegend, clone 53-6.7), FoxP3 (1:200; eBioscience, clone FJK-16s), and Ly6G (1:200; eBioscience, clone 1A8). Samples were stained overnight at 4°C, washed in PBS, labelled with secondary antibodies (all 1:500; AlexaFluor488, A-21208; AlexaFluor555, A-31572; AlexaFluor555, A-21434; AlexaFluor647, A-21247; Thermo Fisher) for 1 hr at room temperature and stained for 10 min with 4,6-diamidino-2-phenylindole (DAPI; 1 mg/ml stock solution diluted 1:50,000 in PBS; D1306, Thermo Fisher) at room temperature with at least four PBS washes in-between. For horizontal wholemounts, 60 µm sections were immunostained as described previously (*Rognoni et al., 2016*). Thin sections were mounted with ProLong Gold Antifade Mountant (Thermo Fisher), while horizontal wholemounts were mounted with glycerol.

For human tissue fibroblast staining, cryosections of 7 µm thickness were fixed with 4% paraformaldehyde/PBS (20 min at room temperature), blocked with 2.5% normal horse serum/TBS (20 min at room temperature), and then incubated overnight with the following primary antibodies: CD39

(eBioscience, clone eBioA1 [A1], in blocking solution; 1:200) and VM (Cell Signaling, #5741; 1:500). After three TBS washes, sections were incubated for 30 min with anti-mouse A488 and anti-rabbit A594 secondary antibodies (VectorFluor Dylight Duet kit; DK-8828). Thereafter sections were incubated with DAPI, washed, and mounted. Counting of papillary fibroblasts (CD39+, VM+ cells) was performed in three skin sections per biopsy with three images per section and averages were calculated. Due to large variations in fibroblast density between donors, the data were normalised to the non-UVR-exposed skin sample of each donor.

For the EP4 immunostaining in human skin, paraffin sections of 5 µm thickness were rehydrated, permeabilised with 0.5% Triton X-100/TBS (10 min), and endogenous hydrogen peroxide activity was blocked using 0.3% hydrogen peroxide/PBS (10 min). After blocking with 2.5% normal horse serum/TBS, tissue sections were incubated with rabbit polyclonal EP4 (1:50; Cat# 101775, Cayman) for 1 hr at room temperature. Primary antibody binding was visualised using a anti-rabbit Vector ImmPress kit (MP-5401, Vector Labs) according to the manufacturer's instructions, and sections counterstained with nuclear fast red (Vector Labs). Images were acquired using the 3D Histech Pannoramic 250 Flash II slide scanner with a ×20/0.80 Plan Apo objective. EP4 expression was analysed in three skin sections (three fields of view per section) per time point using ImageJ software (NIH, UK); thresholding was used to mask positively stained areas, and the percentage area of epidermis or dermis was calculated.

For collagen hybridising peptide (CHP) staining (*Hwang et al., 2017*), 14 µm cryosections of back skin were fixed with 4% paraformaldehyde/PBS (10 min at room temperature), permeabilised with 0.1% Triton X-100/PBS (10 min at room temperature), blocked with 5% BSA/PBS (1 hr at room temperature), and stained with the indicated primary antibodies and 5 µM B-CHP (BIO300, 3Helix) overnight at 4°C. According to the manufacturer's instructions, the B-CHP probe was heated for 5 min at 80°C before adding it to the primary antibody mixture, which was immediately applied to the tissue sections. Sections were washed four times with PBS and incubated with appropriate secondary antibody and streptavidin–AlexaFluor647 (S32357, Thermo Fisher) for 1 hr at room temperature. After an additional four washes and DAPI staining slides were mounted as described above.

Confocal microscopy was performed with a Nikon A1 confocal microscope using a ×20 objective, and brightfield images of H&E and Herovici staining were acquired using a Hamamatsu digital slide scanner with a ×40 objective.

## In vivo live imaging and analysis

In vivo live imaging of dermal fibroblasts was performed after 1 and 4 days of acUVB exposure (see experimental design in *Figure 4A*). Briefly, prior to skin imaging hair follicles were removed with depilation cream (Veet hair removal cream for dry skin) which was applied to the sham and UVR-exposed skin area of the lower back and massaged into the skin for approximately 2 min. The area was then washed thoroughly with water, removing hair and cream from the imaging site. Throughout imaging, mice were anaesthetised by inhalation of vaporised 1.5% isoflurane (Cp-Pharma) and placed in the prone position in a chamber with body temperature maintained at 37°C via a homeothermic monitoring system (Harvard Apparatus). Additionally, oxygen levels were monitored with the MouseOx Plus (Starr Life Sciences Corp) throughout the imaging sessions using an adult mouse pinch attached to the thigh. Oxygen saturation remained at approximately 99%.

The back skin was stabilised between a cover glass and a thermal conductive soft silicon sheet as previously described (*Hiratsuka et al., 2015*). Two-photon excitation microscopy was performed with a Zeiss LSM 7MP upright microscope, equipped with a W Plan-APOCHROMAT ×20/1.0 water-immersion objective lens (Zeiss) and a Ti:Sapphire laser (0.95 W at 900 nm; Coherent Chameleon II laser). The laser power used for observation was 2–10%. Scan speed was 4 ls/pixel. The nuclear GFP expression of dermal fibroblasts can be readily detected in PDGFRαH2BEGFP transgenic mice, and the autofluorescence of fibrillar collagen can be visualised by the second harmonic generation (SHG) using an excitation wavelength of 770 nm. For time-lapse images, z-stacks were acquired every 10 min with a view field of 0.257 mm$^2$ in 5 µm steps. A total of 3–6 mice per time point were examined, and the duration of time-lapse imaging was 70–90 min per mouse. Optimisation of image acquisition was performed to avoid fluorescence bleaching and tissue damage and to obtain the best spatiotemporal resolution. Acquired images were analysed with Fiji imaging software (ImageJ, NIH) and Imaris (BitPlane).

Briefly, raw image files (czi) were imported into Fiji where they were subjected to the Correct 3D Drift plugin using channel 1 (collagen) for registration and selection for sub-pixel drift correction (*Parslow et al., 2014*). The sample drift correction was then manually checked via orthogonal view whereby three hair follicles in each sample were selected in the xz and yz planes and their relative positions measured at time 0 and 80 min and an average taken. The final 3D drift-corrected time-lapse movies were then inputted into Imaris (BitPlane). Within Imaris, tracking spots over time were selected using channel 2 (GFP) to follow fibroblast movement over time. Only fibroblasts with a signal quality above 80%, diameter above 8 µm, and max distance 15 µm were selected using the autoregressive motion algorithm. This created an animation with spots corresponding to fibroblasts. All spots which corresponded to epidermal noise or hair follicle signal were removed manually from each image. Statistics, including cell displacement, position, and mean velocity, were exported into Excel for further analysis. For calculation of the cell displacement in x-, y-, and z-directions, the position of each cell at the start and imaging endpoint was compared and the percentage of cells displaced ≥5 µm (z-stack imaging step size) was quantified.

## Prostaglandin E2 measurement

The prostaglandin E2 (PGE-2) content of skin samples was determined by using a Prostaglandin E2 ELISA Kit (Cayman Chemicals). Briefly, a tissue piece (<80 mg, preserved in OCT at –80°C) was homogenised in 1 ml of lysis buffer (ELISA Buffer supplemented with 10 µM indomethacin) with a gentleMACS dissociator (Miltenyi Biotec). After a centrifugation step ($8000 \times g$, 10 min), PGE-2 was assayed in the supernatant (1:120 dilution with ELISA Buffer) by following the manufacturer's protocol. The concentrations were calculated using a standard curve of PGE-2 between 15 and 500 pg/ml.

## Single-cell RNA-seq analysis of neonatal and adult fibroblast populations and human transcriptomic data

Single-cell RNA-seq data from neonatal (P2) and adult (P21) mouse skin were obtained from GEO dataset GSE153596 (*Phan et al., 2020*). Pre-processing and initial data analysis were performed with Scanpy (*Wolf et al., 2018*) according to the tutorial provided (https://scanpy-tutorials.readthedocs.io/en/latest/pbmc3k.html). Briefly, data were imported, cells expressing less than 200 genes and genes expressed in less than 3 cells were filtered out, counts were normalised, logarithmised, and scaled, a PCA and UMAP were computed, and cells were clustered using the Leiden algorithm. The fibroblast subset was selected and clustered and annotated using marker genes as previously described (*Phan et al., 2020*). Finally, GO term enrichment using g:Profiler (*Raudvere et al., 2019*) was performed on differentially increased genes (p<0.05, Log2 fold change > 1) for each cluster.

Human single-cell RNA-seq data from eyelid skin was visualised using the single-cell ageing atlas (*Zou et al., 2021*; *Liu et al., 2021*), and GO term enrichment of significant genes (Log2 fold change > 0.5, p<0.05) in each identified fibroblast cluster was performed with g:Profiler (*Raudvere et al., 2019*). In addition, human RNA-seq data for upper and lower breast dermis from three donors were obtained from GSE109822 (*Philippeos et al., 2018*), and differentially expressed genes were determined with edgeR (*Robinson et al., 2009*). GO term enrichment was performed on differentially increased genes (Log2 fold change > 0.5, p<0.05) using g:Profiler (*Raudvere et al., 2019*).

## Quantitation and statistical analysis

Statistical analysis was performed with GraphPad Prism 9 software. Unless stated otherwise, data are means ± standard deviation (SD) and statistical significance was determined by unpaired *t*-test, ordinary one-way or two-way ANOVA for biological effects with an assumed normal distribution. For unbiased cell identification with DAPI, Ki67, YAP, yH2AX, cCasp3, TOPGFP, or PDGFRαH2BEGFP labelling, nuclear staining was quantified using the Spot detector plugin of Icy software (version 2.1.0.1). Similarly, cells labelled with tdTomato in the lineage-tracing experiments or stained with COX-2, Ly6G (neutrophils), CD3 (CD3+ T cells), CD8 (cytotoxic T cells), and FoxP3+ (Tregs) were counted per area with the Spot detector plugin of Icy software. To quantify CD45, EP4, and CHP staining in tissue sections, mean fluorescence was determined with Icy software and normalised to background. Cell elongation was determined by measuring the maximal tdTomato+ cytoplasm diameter of individual cells in the papillary dermis of Lrig1-CreER × Rosa26-tdTomato lineage-traced transgenics. Figures were prepared with Adobe Photoshop and Adobe Illustrator (CC2021).

## Acknowledgements

We thank the Nikon Imaging Centre, Dylan Herzog from the Microscopy Innovation Centre here, and BSU facility at KCL for expert assistance. Further, we would like to thank Matteo Battilocchi (KCL) and Dr Monica Sen (KCL) for assistance with in vivo experiments and the lymph node isolation, respectively. We would also like to thank Prof Edel O'Toole (Queen Mary University of London) for critically reading the manuscript. FMW acknowledges financial support from the UK Medical Research Council (MR/PO18823/1), the Wellcome Trust (206439/Z/17/Z) and Cancer Research UK (C219/A23522). ER is the recipient of a European Molecular Biology Organization (EMBO) long-term fellowship (ALTF594-2014) and advanced fellowship (ALTF523-2017), and TK received funding from the Medical College of Saint Bartholomew's Hospital Trust. This work was funded by grants to FMW. The human study was funded by the Wellcome Trust (grant WT94028, LER) and the NIHR Manchester Biomedical Research Centre (NJH, LER).

The authors acknowledge the use of core facilities provided by financial support from the Department of Health via the National Institute for Health Research (NIHR) comprehensive Biomedical Research Centre award to Guy's & St Thomas' NHS Foundation Trust in partnership with King's College London and King's College Hospital NHS Foundation Trust.

## Additional information

### Competing interests

Fiona M Watt: FW is on secondment as Executive Chair of the Medical Research Council. The author has no other competing interests to declare. The other authors declare that no competing interests exist.

### Funding

| Funder | Grant reference number | Author |
| --- | --- | --- |
| Cancer Research UK | C219/A23522 | Fiona M Watt |
| Medical Research Council | MR/PO18823/1 | Fiona M Watt |
| Wellcome Trust | 206439/Z/17/Z | Fiona M Watt |
| Wellcome Trust | WT94028 | Lesley E Rhodes |
| NIHR Manchester Biomedical Research Centre | | Nathan J Hawkshaw Lesley E Rhodes |
| European Molecular Biology Organization | ALTF594-2014 | Emanuel Rognoni |
| Medical College of Saint Bartholomew's Hospital Trust | | Thomas Kirk |

The funders had no role in study design, data collection and interpretation, or the decision to submit the work for publication.

### Author contributions

Emanuel Rognoni, Conceptualization, Data curation, Investigation, Methodology, Project administration, Validation, Visualization, Writing – original draft, Writing – review and editing; Georgina Goss, Kalle H Sipilä, Investigation, Methodology, Writing – review and editing; Toru Hiratsuka, Niwa Ali, Methodology; Thomas Kirk, Investigation, Methodology; Katharina I Kober, Prudence PokWai Lui, Victoria SK Tsang, Nathan J Hawkshaw, Suzanne M Pilkington, Inchul Cho, Investigation; Lesley E Rhodes, Methodology, Resources, Writing – review and editing; Fiona M Watt, Funding acquisition, Resources, Supervision, Writing – original draft, Writing – review and editing

### Author ORCIDs

Emanuel Rognoni (iD) http://orcid.org/0000-0001-6050-2860

Toru Hiratsuka (iD) http://orcid.org/0000-0002-5359-2690
Katharina I Kober (iD) http://orcid.org/0000-0002-8076-3379
Inchul Cho (iD) http://orcid.org/0000-0001-5527-0962
Niwa Ali (iD) http://orcid.org/0000-0003-4473-8747
Lesley E Rhodes (iD) http://orcid.org/0000-0002-9107-6654
Fiona M Watt (iD) http://orcid.org/0000-0001-9151-5154

## Ethics

Human subjects: Ethical approval was granted by the Greater Manchester North NHS research ethics committee (ref: 11/NW/0567) for the studies presented in Figure 1 and Figure 6. Details of the time course analysis of UVR challenged human skin have been reported previously (Hawkshaw NJ et al. 2020). All volunteers provided written informed consent in accordance with the Declaration of Helsinki principles.

All animal experiments were subject to local ethical approval and performed under the terms of a UK government Home Office license (PPL 70/8474 or PP0313918).

## Decision letter and Author response

Decision letter https://doi.org/10.7554/eLife.71052.sa1
Author response https://doi.org/10.7554/eLife.71052.sa2

## Additional files

### Supplementary files
- Supplementary file 1. Immune cell flow cytometry antibody panel for T cells and myeloid cells.

- Transparent reporting form

### Data availability
All data generated or analysed during this study are included in the manuscript and supporting files. Source data files containing the numerical data used to generate the figures have been provided for all figures.

The following previously published datasets were used:

| Author(s) | Year | Dataset title | Dataset URL | Database and Identifier |
|---|---|---|---|---|
| Phan QM, Fine G, Salz L, Herrera GG, Wildman B, Driskell IM, Driskell RR | 2020 | Lef1 expression in fibroblasts maintains developmental potential in adult skin to regenerate wounds | http://www.ncbi.nlm.nih.gov/geo/query/acc.cgi?acc=GSE153596 | NCBI Gene Expression Omnibus, GSE153596 |
| Philippeos C, Telerman SB, Oulès B, Pisco AO, Shaw TJ, Elgueta R, Lombardi G, Driskell RR, Soldin M, Lynch MD, Watt FM | 2018 | Spatial and single-cell transcriptional profiling identifies functionally distinct human dermal fibroblast subpopulations | https://www.ncbi.nlm.nih.gov/geo/query/acc.cgi?acc=GSE109822 | NCBI Gene Expression Omnibus, GSE109822 |
| Zou Z, Long X, Zhao Q, Zheng Y, Song M, Ma S, Jing Y, Wang S, He Y, Esteban CR, Yu N, Huang J, Chan P, Chen T, Izpisua Belmonte JC, Zhang W, Qu J, Liu GH | 2020 | A Single-Cell Transcriptomic Atlas of Human Skin Aging | https://ngdc.cncb.ac.cn/gsa-human/browse/HRA000395 | GSA:HRA000395, HRA000395 |

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

## Appendix 1

### Appendix 1—key resources table

| Reagent type (species) or resource | Designation | Source or reference | Identifiers | Additional information |
|---|---|---|---|---|
| Strain, strain background (*Mus musculus*) | PDGFRα H2BEGFP, C57Bl6/BalbC | PMID:12748302 | RRID:MGI:2663656 | |
| Strain, strain background (*M. musculus*) | *Lrig1*-CreERt2-IRES-GFP, C57Bl6/BalbC | PMID:23954751 | RRID:MGI:5520983 | |
| Strain, strain background (*M. musculus*) | *Dlk1*-CreERt2, C57Bl6/BalbC | PMID:24336287 | RRID:MGI:5555961 | |
| Strain, strain background (*M. musculus*) | ROSAfl-stopfl-tdTomato, C57Bl6/BalbC | Jackson Laboratories | Stock no.:007905 | |
| Strain, strain background (*M. musculus*) | *Krt14*ΔNβ-cateninER, C57Bl6/BalbC | PMID:15084463 | RRID:MGI:6315261 | |
| Strain, strain background (*M. musculus*) | TCF/Lef:H2B-GFP, C57Bl6/BalbC | PMID:21176145 | RRID:MGI:4881498 | |
| Antibody | Anti-vimentin (rabbit polyclonal) | Cell Signaling | Cat# 5741 | IF (1:500) |
| Antibody | Anti-Ly6G (rat monoclonal) | eBioscience | Clone 1A8 | IF (1:200) |
| Antibody | Anti-FoxP3 (rat monoclonal) | eBioscience | Clone FJK-16s | IF (1:200) |
| Antibody | Anti-CD8 (rat monoclonal) | BioLegend | Clone 53-6.7 | IF (1:200) |
| Antibody | Anti-CD3 (rat monoclonal) | BioLegend | Clone 17A2 | IF (1:200) |
| Antibody | Anti-CD49f (rat monoclonal) | BioLegend | Clone GoH3 | IF (1:500) |
| Antibody | Anti-CD45 (rat monoclonal) | eBioscience | Clone 30-F11 | IF: (1:200) |
| Antibody | Anti-CD31 (rat monoclonal) | eBioscience | Clone 390 | IF: (1:200) |
| Antibody | Anti-COX-2 (rabbit polyclonal) | Abcam | Cat# ab15191 | IF (1:500) |
| Antibody | Anti-cCasp3 (rabbit polyclonal) | Cell Signaling | Cat# 9661 | IF (1:500) |
| Antibody | Anti-YAP (rabbit polyclonal) | Cell Signaling | Cat# 14074 | IF (1:500) |
| Antibody | Anti-EP4 (rabbit polyclonal) | Bioss | Cat# BS-8538R | IF (1:200) |
| Antibody | Anti-EP4 (rabbit polyclonal) | Cayman | Cat# 101775 | IF (1:50) |
| Antibody | Anti-Ki67 (rabbit polyclonal) | Abcam | Cat# ab16667 | IF (1:500) |
| Antibody | Anti-Ki67 (rat monoclonal) | Invitrogen | Clone SolA15 | IF (1:500) |
| Antibody | Anti-yH2AX (rabbit polyclonal) | Abcam | Cat# ab81299 | IF (1:500) |
| Antibody | Anti-αSma (rabbit polyclonal) | Abcam | Cat# ab5694 | IF (1:500) |
| Antibody | Anti-human CD39 (mouse monoclonal) | eBioscience | Clone eBioA1 (A1) | IF (1:200) |
| Antibody | Anti-mouse CD4 (rat monoclonal) | BioXCell | Clone GK1.5 | For immune cell depletion |
| Antibody | Anti-mouse CD8 (rat monoclonal) | BioXCell | Clone 2.43 | For immune cell depletion |
| Antibody | Anti-IgG (rat monoclonal) | BioXCell | Clone LTF-2 | For immune cell depletion |
| Antibody | Anti-rat AlexaFluor488 (donkey polyclonal) | Thermo Fisher | Cat# A-21208 | IF (1:1000) |
| Antibody | Anti-rabbit AlexaFluor555 (donkey polyclonal) | Thermo Fisher | Cat# A-31572 | IF (1:1000) |
| Antibody | Anti-rat AlexaFluor555 (goat polyclonal) | Thermo Fisher | Cat# A-21434 | IF (1:1000) |

*Appendix 1 Continued on next page*

*Appendix 1 Continued*

| Reagent type (species) or resource | Designation | Source or reference | Identifiers | Additional information |
|---|---|---|---|---|
| Antibody | Anti-rat AlexaFluor647 (goat polyclonal) | Thermo Fisher | Cat# A-21247 | IF (1:1000) |
| Antibody | Anti-mouse/rat Foxp3 eFluor450 (rat monoclonal) | eBioscience | Clone FJK-16s | FACS (1:100) |
| Antibody | Anti-mouse CD152 (CTLA4), PE (rat monoclonal) | BD | Clone UC10-4F10-11 | FACS (1:100) |
| Antibody | Anti-human Ki67, PE-Cy7 (mouse monoclonal) | BD | Clone B56 | FACS (1:100) |
| Antibody | Anti-mouse TCR gd, PerCP-Cy 5.5 (rat monoclonal) | BioLegend | Clone GL3 | FACS (1:300) |
| Antibody | Anti-mouse CD45, Alexa Fluor700 (rat monoclonal) | eBioscience | Clone 30-F11 | FACS (1:200) |
| Antibody | Anti-mouse CD25, APC-eFluor780 (rat monoclonal) | eBioscience | Clone PC61.5 | FACS (1:150) |
| Antibody | Anti-mouse CD8a, Brilliant Violet 605 (rat monoclonal) | BioLegend | Clone 53-6.7 | FACS (1:200) |
| Antibody | Anti-mouse CD4 Antibody, Brilliant Violet 650 (rat monoclonal) | BioLegend | Clone RM4-5 | FACS (1:200) |
| Antibody | Anti-mouse CD3, Brilliant Violet 711 (rat monoclonal) | BioLegend | Clone 17A2 | FACS (1:150) |
| Antibody | Anti-mouse CD11b, AlexaFluor647 (rat monoclonal) | BioLegend | Clone M1/70 | FACS (1:400) |
| Antibody | Anti-F4/80, PE-Cy5 (rat monoclonal) | eBioscience | Clone BM8 | FACS (1:400) |
| Antibody | Anti-MHC Class II (I-A) (NIMR-4), PE (rat monoclonal) | eBioscience | Clone M5/114.15.2 | FACS (1:500) |
| Antibody | Anti-mouse Ly-6A/E (Sca-1), APC/Cy7 (rat monoclonal) | BioLegend | Clone D7 | FACS (1:400) |
| Antibody | Anti-mouse CD11c, AlexaFluor488 (hamster monoclonal) | BioLegend | Clone N418 | FACS (1:400) |
| Peptide, recombinant protein | B-CHP | 3Helix | Cat# BIO300 | IF (1:100) |
| Peptide, recombinant protein | Streptavidin–AlexaFluor647 | Thermo Fisher | Cat# S32357 | IF (1:500) |
| Peptide, recombinant protein | Collagenase XI | Sigma-Aldrich | Cat# C7657 | |
| Peptide, recombinant protein | Hyaluronidase | Sigma-Aldrich | Cat# H4272 | |
| Peptide, recombinant protein | DNase I | Sigma-Aldrich | Cat# DN-25 | |
| Other | 4,6-Diamidino-2-phenylindole (DAPI) | Thermo Fisher | Cat# D1306 | IF (1 mg/ml stock solution diluted 1:50,000) |
| Other | Ghost Dye Violet 510 Live/Dead Stain | Tonbo Biosciences | Cat# 13-0870T100 | FACS (1:500) |
| Other | UVR system (Tyler Research UV-2) | Tyler Research | Cat# UV-2 | |
| Commercial assay or kit | VectorFluor Dylight Duet kit | Vector Labs | Cat# DK-8828 | |

*Appendix 1 Continued on next page*

*Appendix 1 Continued*

| Reagent type (species) or resource | Designation | Source or reference | Identifiers | Additional information |
|---|---|---|---|---|
| Commercial assay or kit | Anti-rabbit Vector ImmPress kit | Vector Labs | Cat# MP-5401 | |
| Commercial assay or kit | Prostaglandin E2 ELISA Kit | Cayman Chemicals | Cat# 500141 | |
| Commercial assay or kit | Celecoxib | Selleckchem | Cat# S1261 | |
| Commercial assay or kit | UltraComp eBeads | Thermo Fisher | Cat# 01-2222-42 | |
| Commercial assay or kit | ArC Amine Reactive Compensation Bead Kit | Thermo Fisher | Cat# A10346 | |
| Commercial assay or kit | 4-Hydroxyt amoxifen (4OHT) | Sigma-Aldrich | Cat# H7904 | |
| Commercial assay or kit | Tamoxifen | Sigma-Aldrich | Cat# T5648 | |
| Software, algorithm | ICY | Institut Pasteur and France-BioImaging | RRID:SCR_010587 | |
| Software, algorithm | Fiji imaging software (ImageJ) | NIH | RRID:SCR_002285 | |
| Software, algorithm | Imaris | BitPlane | RRID:SCR_007370 | |
| Software, algorithm | Scanpy | Scanpy | RRID:SCR_018139 | |
| Software, algorithm | g:Profiler | g:Profiler | RRID:SCR_006809 | |
| Software, algorithm | FlowJo | FlowJo | RRID:SCR_008520 | |
| Software, algorithm | GraphPad Prism | GraphPad Prism | RRID:SCR_002798 | |

