## [Editor Report]

Your study adds important and novel information regarding how the skin responds to UV radiation and the subsequent repair and regenerative response.

---

## [Decision Letter]

[Editors' note: this paper was reviewed by Review Commons.]

---

## [Author Response]

Reviewer #1 (Evidence, reproducibility and clarity (Required)):Summary:In this paper, Rognoni et al. showed that distinct fibroblast population (papillary fibroblast in the upper dermis vs reticular fibroblast in the lower dermis) in the skin have different response to UV radiation and tissue remodeling. Taking advantage of specific CreER tools that they have established in previous work, they nicely demonstrated that papillary fibroblasts are the cell type, which is responsible for the contribution of tissue repair after acute UV radiation. In contrast, chronic UV-induced damage is not easily repaired in part due to more severe loss of papillary fibroblasts, which is also seen in aging skin. In the last part, the authors tried to addressed further the molecular mechanisms and showed that the CD4 and CD8 T cells have a supportive role in UV-induced tissue repair by maintaining the papillary fibroblasts.Major comments:I think minimal work is required for the submission before publication in a peer reviewed journal.

We thank the reviewer for this positive evaluation of our manuscript.

I recommend authors to explain a bit more about the possible factors to explain the cell response differences between papillary vs reticular fibroblasts. For me, first part (fibroblast heterogeneity) and second part (Wnt, interaction with immune cells) do not connect so well.

To explore potential intrinsic differences between papillary and reticular fibroblasts, we have included an analysis of GO terms associated with the cellular response to UV, cell stress and DNA damage/repair in publicly available transcriptomic datasets and how these map to differences in the gene expression profiles of different fibroblasts subpopulations in undamaged neonatal and adult mouse skin (Phan et al., 2020, e*Life*), sun exposed human eyelid skin (Zou et al., 2021, Dev Cell) as well as microdissected human skin (Philippeos et al., 2018, J Invest Dermatol.) (Figure 1—figure supplement 2). Our new data analysis indicates that dermal fibroblast subpopulations do not differ in their response or susceptibility to UV damage.

We have improved the connection between the two parts of the manuscript by re-writing the text and by expanding our analysis of Wnt signalling (Figure 5) and immune cell distribution (Figure 6 and Figure 6—figure supplement 1) in the upper and lower dermis.

Are there any differences in immune cell distribution (or EP4 expression) between upper and lower dermis? Or not, what makes differences between papillary vs reticular fibroblasts in their response to UV? The CD4 and CD8 T cells seem to act mainly on papillary fibroblasts based on the blocking antibody effects as shown in Figure 6C. Are CD4 and CD8 T cells predominantly recruited and located in the upper dermis? The authors should provide quantification of CD8 T cell distribution in upper vs lower dermis (Supplementary Figure 6), like they did in other figures.

We have now quantified the distribution of different T cell populations in the upper and lower dermis and included the data in Figure 6A-C and Figure 6—figure supplement 1C. While CD3+ T cells were depleted in the epidermis, they were increased in the upper and lower dermis. Cytotoxic T cells (CD8^+^), which are enriched in the lower dermis in control skin, were significantly increased in the upper dermis after UVR exposure while Tregs (FoxP3+) significantly expanded throughout the dermis.

In addition, we have now measured prostaglandin E2 (PGE-2) concentration after UVR exposure with and without T cell depletion (Figure 7A and B) and expanded our analysis of EP4 and COX-2 expression in the upper and lower dermis (Figure 7C and K). Acute UVR exposure induced the release of PGE-2 in the skin, which was further increased after CD4^+^ and CD8^+^ cell depletion. COX-2 and EP4 expression are highly increased 1 day after UVR exposure in the epidermis and dermis, coinciding with the fibroblast loss.

Minor comments:The Figure overall is presented very well, and writing is clear. I am a bit confused with different time points presented in each panel of figures. Most of places in their figures, they labeled clearly e.g., “acUVB π 1 day”, but not all. For example, Figure 2A is 8 weeks post UVB, Figure 2B, chUVB is 1 or 3 day. Figure 2E is 24 hours. How about Figure 2C, D? I appreciate if authors could use the same labeling across all figures, which really help readers to understand the time course of cellular events upon UVR.

We apologize for the inconsistency, and we have now made the figure labels clearer and more consistent throughout the manuscript.

In Figure 6D, aCD4 and aCD8, "a" (alpha) is not written in symbol.

We have corrected this.

Reviewer #1 (Significance (Required)):Significance:The significance of this paper is to describe a distinct cellular response of heterogeneous dermal fibroblasts in the context of UV-induced tissue damage. They used unique mouse models to address this point, which has not been shown in the field. This study is relevant for broad area of skin researchers including the field of aging, wound healing, tissue remodeling, cancer and immune-dermal interaction. I am expertise in skin stem cell biology and particularly interested in the heterogeneity of stem cells.

We thank the reviewer for acknowledging the novelty and significance of our work.

Referee Cross-commenting:I have read the comments from referees 2 and 3. I agree with their suggestions. Especially, some additional clarification and experiments on the UVB radiation, live imaging, immune cell response will improve the clarity of the manuscript.

Please see responses to the other reviewers.

Reviewer #2 (Evidence, reproducibility and clarity (Required)):In this manuscript, Rognoni and coworkers analyze the response of dermal fibroblasts to acute and chronic UV exposure. They show that both in human and mouse skin, acute UV leads to transient depletion of papillary fibroblasts characterized by increased gH2AX signal and apoptosis, followed by a wave of increased proliferation, whereas no substantial ECM remodeling is seen. They further show that chronic UV leads to a more stable decrease in papillary fibroblasts, but less apoptosis and compensatory proliferation are observed. In contrast substantial ECM remodeling is seen. In addition chronic UV-exposed fibroblasts appear elongated. The authors then show that acute UV is able to trigger some fibroblast motility, whereas boosting proliferation is not sufficient to rescue fibroblast numbers. In contrast, blunting inflammation results to some degree of rescue in terms of fibroblast numbers.Overall this is an interesting manuscript that utilizes stringent approaches to address the clinically relevant question of effects of UV on skin. The manuscript is very well and clearly written, the data is of high quality and compelling. The conclusions are overall well justified by the data.

We thank the reviewer for these comments.

The only major issue I have with the manuscript is that it is not clear how the in vivo imaging, a very elegant approach with a compelling result, integrates with the rest of the data. From the first part it seems that the authors draw the conclusion that acute UV kills the fibroblasts through DNA damage and apoptosis, which is not seen in the chronic treatment. So it would have been more logical to analyse the migration patterns in the chronic UV group as migration would be more likely to be relevant there. Also, if cells are migrating downwards, should this not result in increased levels of Lrig1+ cells in the lower dermis. Now both compartments seem to show a reduction (Figure 3d). On the other hand, if the migration would be replenishing the lost population, then one would expect the migration to be directional or at least show some biased z displacement towards the epidermis, which seems not to be the case. So all in all the reader is left wondering about the relevance of this observation. While I understand that repeating the imaging in chronic UV treated mice is a major experimental undertaking, this or additional experiments addressing the role of migration in the described phenomena would strengthen this manuscript.

We carried out in vivo live imaging of dermal fibroblasts to elucidate firstly if active cell migration contributes to fibroblast depletion in the upper dermis (which occurs in both acute and chronic UV exposed skin), and secondly, how fibroblast migration is involved in the early tissue damage repair response (repopulation of the upper dermis only occurs in after acute UV damage). Our conclusion is that after UVR exposure there is very little cell migration and that it is random, not directional. This is in contrast to fibroblast behaviour during repair of full thickness skin wounds (Rognoni et al., 2018, Mol Syst Biol; Jiang et al., 2020, Nat Commun). We conclude that following UVR (acute or chronic) fibroblasts are depleted through cell death and not through active downward migration. This is in agreement with our Lrig1CreER lineage tracing data, which shows no fibroblast density increase in the lower dermis. During the early tissue repair response (4 days post-UVB exposure) fibroblasts become more motile, which correlates with the observed reorganization of papillary lineage fibroblasts. However, the lack of directionality suggests that the replenishment of upper dermal fibroblasts is a stochastic process, similar to the reorganisation of fibroblast lineages we previously observed during skin ageing (Rognoni et al., 2018, Mol Syst Biol).

We have now revised the manuscript (Results and Discussion) to improve the description of our findings and state more clearly that fibroblast loss is not due to migration.

Another issue is that the matrix remodeling and cell elongation phenomena remain slightly isolated observations and it is not clear how they tie to the other observed changes.

We have clarified this in the text by pointing out the links between fibroblast migration, cell shape changes and ECM remodelling. We think it is particularly interesting that in chronic UVR exposed skin the reduction in fibroblast density is partly compensated by elongation of upper dermal fibroblasts. This phenomenon is also observed in aged skin (Marsh et al., 2018, Cell) and our data indicate that repeated UVB tissue damage – which occurs in photo-ageing – accelerates the process.

Minor points:1. The images in Figure 3D for the Dlk1 Cre labeling are difficult to interpret as there are very few labeled cells to begin with. This is in stark contrast to Figure 3B where a large number of cells are visible. Yet the quantified Dlk1+ cell densities in 3D and 3B are similar. Are these images not representative or what would explain this difference?

We have replaced the image in Figure 3D with one that is a better representation of the quantification. We have also made the quantification of Dlk1Cre labelled cells in upper and lower dermis consistent.

2. Cell elongation statistical analyses in 3F would be more robustly done from means individual mice rather than pooling all cells together thus inflating the n number.

We have now included this analysis in Figure 3—figure supplement 1 C and D.

3. Overall it would be more compelling to show individual replicates than bar graphs throughout the manuscript.

We agree and have revised the Figures accordingly.

Reviewer #2 (Significance (Required)):Overall this is an interesting, high quality manuscript that provides compelling new knowledge on the effects of UV irradiation on the skin. The results will be interesting for the fields of skin biology, DNA damage and UV.Reviewer #3 (Evidence, reproducibility and clarity (Required)):This reviewer has expertise in epidermal biology clinical dermatology and has worked in the context of UV irradiation and carcinogenesis.In this work, authors explore the effect of acute and chronic UV on dermal fibroblast populations. Making use of murine reporter lines and lineage tracing they show the partial loss of a specific population of fibroblasts. Mechanistically, the role of immune cells and prostaglandins are explored. Finally, to some extent, the findings are associated with similar observations in human samples obtained from a small clinical trial.

Major comments:This is a comprehensive study claiming many novel findings.

We thank the reviewer for this positive evaluation of our study.

1. UVA is the main component of Solar UV and is the main type of UV penetrating the dermis and acting on both the cells and the extra-cellular matrix through oxidative stress. It is unclear why radiation in these experiments was restricted to UVB.

While we are aware of the importance of UVA for skin photobiology, it is evident from the literature that UVA and UVB induce different types of photo-damage in skin and so a separate analysis of UVA and UVB provides useful mechanistic insights that could lead to design of specific interventions. In regard to environmental risk factors there is an increasing interest in the specific impact of UVB radiation on human health, as ambient UVB will decrease or increase depending on the success of the Montreal Treaty in limiting ozone destruction (see recent United Nations EEAP 2019 report: Environmental Effects and Interactions of Stratospheric Ozone Depletion, UV Radiation, and Climate Change, https://ozone.unep.org/science/assessment/eeap). We have included these points in the revised manuscript.

2. The main issue relates to the definition of upper dermis versus lower dermis fibroblasts. The anatomical separation is not clarified in this paper. Moreover, at the molecular level, there seems to be a gap between DLK1Cre labelled cells and Lrig1Cre labelled cells. This is an important conceptual point as it needs to be clarified if the depletion concerns all cells of the upper dermis lineage or alternatively it only concerns the cells (regardless of biology) that are closest to the surface and within reach of UVB radiation.

We and others have extensively characterized the development and distribution of different dermal fibroblast lineages in mouse skin (see Rognoni et al., 2016, Development; Rognoni et al., 2018, Mol Syst Bio; Driskell et al., 2013, Nature; Rinkevich et al., 2015, Science; Salzer et al., 2018, Cell). In adult mouse skin papillary fibroblasts lie in the upper dermis that extends from the basement membrane to the hair follicle infundibulum above the sebaceous glands. The lower dermis, containing reticular fibroblasts, extends from below the papillary dermis to the DWAT (dermal white adipose tissue; hypodermis). We have indicated the boundary between upper and lower dermis with a dotted line in the histology/immunofluorescence images. Our pan-fibroblast (PDGFRaH2BEGFP+) and Lrig1-CreER labeled dermal cell density quantifications show that one day after acute UVB exposure all fibroblasts close to the basement membrane in the upper dermis are depleted, which correlates with the distribution of apoptotic and DNA damaged fibroblasts and UVR induced tissue damage. GO term analysis of different dermal fibroblast subpopulations (new Figure 1—figure supplement 2) indicates that these subpopulations do not differ in their response or susceptibility to UV damage. We have stated these findings more clearly in the revised manuscript.

3. Regarding the evolution of the fibroblast depletion, authors claim that the numbers reach baseline levels at 14 days after an acute irradiation. However in the chronic irradiation model, they claim that fibroblasts never recover their initial numbers… In the latter set of experiments, experiments have only been performed for up to 10 days. It is advised to continue observing changes in fibroblast numbers up to 14 and 21 days before such conclusion.

We have now included data on fibroblast density at later time points following chronic UVB exposure (Figure 2H), which shows that upper fibroblast depletion persists for 30 days after chronic UVR. Our observations are in line with previous chronic UVB studies (now cited) reporting a reduced cell density in the papillary dermis even 200 days after the final UVB exposure (Dai et al., 2007, Am J Pathol).

The human equivalent of this experiment is also not convincing. It is not clear how many technical replicates (staining) have been performed for each patient. In any case the overall trend does not support the authors description (with the exception of 1 patient) of a depletion and a recovery. There seems to be a general trend towards increased fibroblasts over 14 days. In that sense Figure 2H may be an overrepresentation of the authors findings.

We have revised the quantitation of UVB treated human skin in Figure 1A and included the missing technical information in the Methods section. Due to the large variation in fibroblast density between donors, we have now normalized the mean fibroblast density to the non-UVR exposed skin sample for each donor and present the normalized mean fibroblast densities in a boxplot. Our data show a significant loss of dermal fibroblasts at 1 day post UVR; this is followed by a transient increase during the repair phase and a return to pre-UVR treatment levels after two weeks.

4. I question the decision by the authors to use a burning dose of UVB for the chronic irradiation experiment? Often to reflect the clinical scenario of chronic irradiation sub-erythemal doses are used and adjusted to the tanning response.

Compared to other recent studies (Bald et al., 2014, Nature; Kunisada et al., 2005, Cancer Res; Ohkumo et al., 2006, Mol Cell Biol.; Dai et al., 2007, Am J Pathol.; Mira Han et al., 2017, Sci Rep.; Meeran et al., 2009, Toxicol Appl Pharmacol), our chronic UVB treatment regime in C57BL/6 mice is rather mild with 800 J/m^2^ twice a week for 8 weeks. The mice develop prominent skin tanning after 8 weeks of treatment but there is no breakdown of the skin barrier, which would be a feature of severe sunburn. We have stated this more clearly in the revised manuscript.

5. The mechanism proposed in this work for the depletion and recovery of the fibroblasts is unclear. Authors show that upper dermal fibroblasts undergo DNA damage and apoptosis (staining impossible to see). They also claim through in vivo imaging that the cells largely move downwards. The latter finding cannot be recapitulated by the Lrig1Cre lineage tracing. Finally, another claim is made about the protective role of T cells (in reducing PGE2?) and the importance of PGE2. These different mechanisms are not linked to each other. Can authors deplete only Tregs? And not all CD4 cells? Can they deplete CD8 or CD4 cells and treat with celecoxib? Can they measure apoptosis levels in the T cell depletion studies?

We apologise for the lack of clarity regarding mechanism, which was also highlighted by Reviewer 2. We show that fibroblasts are lost by cell death and not by downward migration (see response to Reviewer 2). We have now elaborated more clearly the effect of UVB-induced inflammation on dermal fibroblast survival and included quantification of fibroblast apoptosis (Figure 6H) and PGE-2 levels (Figure 7, A and B), both of which increase upon T cell depletion. In addition, we have added quantification of COX-2 and EP4 expression in the upper and lower dermis after UVB exposure, and show that both are highly induced at 1 day post UVR.

We agree that it would be interesting to selectively deplete Tregs; however we show that both CD4^+^ and CD8^+^ T cells equally influence dermal fibroblast survival and proliferation (Figure 6D-I). Similarly, repeating the celecoxib treatment in combination with anti-CD8 and CD4 would, in our opinion, provide minimal further insights to the current COX-2 inhibition experiment in Figure 7D-J.

Minor comments:1. The T cell depletion studies are not controlled properly by isotype Antbody. Similarly the effect of interventions is not tested in normal skin to ensure specificity of the effect to UV irradiation.

We thank the reviewer for this comment. The revised manuscript now shows the isotype antibody control data in Figure 6—figure supplement 1H-K and specifically highlights the interventions we have performed on non-UVB treated skin.

2. Many of the immunostaining panels are difficult to see at such a low magnification. DLK1Cre lineage tracing images are not convincing as large pockets of TdTomato can be identified that does not correspond to cells.

We have improved the clarity and the labeling of the immunostaining panels. It is worth noting that induction of Dlk1Cre during early skin development labels reticular fibroblasts that further differentiate into adipocytes. Therefore, in adults the adipocyte layer contains large number of tdTomato+ cells, a point we make in the revised manuscript. We have exchanged and increased the size of Figure 3D, to better represent the associated quantification and show the structure of the tdTomato positive adipocyte layer.

3. I cannot comment on fibroblast shapes in vivo the CHP staining and the evaluation of the dermis ECM.Reviewer #3 (Significance (Required)):Understanding the consequences of UV radiation on the skin is essential for our knowledge on skin carcinogenesis, skin ageing but also inflammation. Very few studies have proposed a clear mechanism for the effect of UV on fibroblast behaviour. Solar UV in the context of human pathology is the main environmental exposure. It is composed in large majority by UVA which has an essential role in inducing fibroblast senescence and skin photo-ageing. Unfortunately by restricting the study to UVB radiation authors restrict its applicability to the clinical scenario.Although I am not an expert in the dermis, there are notable deficits in relating to past studies in carcinogenesis (Paolo Dotto's group) or ageing (IGF signalling and senescence) without going through an exhaustive list.

We have revised the manuscript to cite the past studies.